# Towards Certainty: Exploiting Monotonicity with Fast Marching Methods to Reduce Predictive Uncertainty

## Abstract

In recent years, neural networks have achieved impressive performance on a wide range of tasks. However, neural networks tend to make overly optimistic predictions about out-of-distribution data. When managing model risks, it is important to know what we do not know. Although there have been many successes in detecting out-of-distribution data, it is unclear how we can extract further information from these uncertain predictions. To address this problem, we propose to use three types of monotonicity by solving a mean-variance optimization problem. The fast marching method is proposed as an efficient solution. We demonstrate, using empirical examples, that it is possible to provide confident bounds for a large portion of uncertain predictions by monotonicity.

## 1 Introduction

Deep neural networks (DNNs) have been widely applied in a variety of applications. For high-risk sectors such as the financial sector, predictive power is not the only factor to consider. It is stressed in the model risk management handbook [1] provided by the Office of the Comptroller of the Currency [c2](OCC) [c3c3] that when machine learning (ML) models are applied, they need to **know what they don't know**. It is possible, for example, for a trading model to perform very well under normal circumstances, but to fail during a financial crisis. To prevent losing money, traders might switch to other strategies if the trading model alerts them to significant changes in the trading environment.

However, in practice, the distribution of the population may differ significantly from that of the training set. For instance, in lending, there is an abundance of data on good applicants, but a very risky applicant with numerous past dues might be extremely rare and have not been seen by the model before. Out-of-distribution (OOD) inputs often pose a challenge to the use of DNNs. Unless carefully managed, DNNs may become overconfident about their predictions, resulting in catastrophic results.

[c4]There has been considerable attention given to this challenge in recent years, and the results are encouraging (Lakshminarayanan et al., 2017; Kardan et al., 2021; Bibas et al., 2021; Liang et al., 2018; Yang et al., 2022; Geng et al., 2020; Zhou et al., 2022; Ovadia et al., 2019). Based on the successful detection of OOD data, we ask the following question: **What can we learn from what we don't know?**

As with black-box ML models, we may be unable to obtain additional information. On the other hand, recent developments in **domain-knowledge-inspired machine learning models** have been highly successful and could be used to provide additional information. In particular, **monotonic machine learning models** has

---

[1]https://www.occ.treas.gov/publications-and-resources/publications/comptrollers-handbook/files/model-risk-management/index-model-risk-management.html.

[c2] *Text added.*

[c3] *Text added.*

[c3]The Office of the Comptroller of the Currency charters, regulates, and supervises all national banks, federal savings associations, and federal branches and agencies of foreign banks. In general, the Office of the Comptroller of the Currency provides regulatory guidance to financial institutions. In their handbook, they provide practitioners with a high-level guide to controlling model risk and validating AI and ML models in financial markets.

[c4] ~~This challenge has received considerable attention in recent years and the results have shown promising~~

been very popular in many areas (Repetto, 2022; Chen & Ye, 2022; Liu et al., 2020; Chen & Ye, 2023). There may be additional information to be gained from monotonic models in the case of uncertain predictions. It has been shown by Chen (2022) that individual monotonicity can provide reliable bounds to uncertain inputs. Here is an example of a simple illustration [c5]in credit scoring. Credit scores are based on information in credit reports to predict credit behavior, including the likelihood of repaying a loan in a timely manner. ML models can be used to predict the probability of default, which are then converted into credit scores. Predictions are based on a number of factors, one of which is the number of past-due payments. Let's suppose that an applicant has ten [c6]past-due payments. Although the model may not be certain of this prediction due to its rarity, it does know that five [c7]past-due payments has already been very risky, so ten [c8]past-due payments cannot be any less risky. Consequently, it should be categorized into risky groups.

There has been significant attention paid to monotonicity in the past (Sill, 1997; Cano et al., 2019; Gupta et al., 2020), since it is not only about conceptual soundness but also about **fairness**. In the case of credit scoring when individual monotonicity is involved, if an applicant has one more [c1]past-due payment, [c2]the model should then predict that the probability of default is increasing based on the conceptual soundness and fairness outlined in the OCC's handbook. Otherwise, the model would be unfair since an additional past-due payment has not been penalized. Thus, monotonicity is usually a **hard requirement** for related applications.

While individual monotonicity has been extensively studied (Liu et al., 2020; Milani Fard et al., 2016; You et al., 2017; Runje & Shankaranarayana, 2023), **pairwise monotonicity** has been largely ignored. Recent studies (Gupta et al., 2020; Cotter et al., 2019; Chen & Ye, 2022; 2023) have shown that pairwise monotonicity is also important. The idea behind pairwise monotonicity is that some features are intrinsically more important than others. For example, in credit scoring, [c3]past-due payment information can be divided into two features based on the number of [c4]past-due payments within three months or more than three months. It is then important to consider the feature that counts the number of [c5]past-due payments of more than three months as more important for fairness. Alternatively, if the ML model predicts an applicant with a [c6]past-due payment within three months is more risky than another with a [c7]past-due payment of more than three months, then the prediction is unfair.

It is possible to provide more information for models containing pairwise monotonicity. For example, if an ML model is sure that an applicant with three [c8]past-due payments less than three months is already very risky, then an applicant with three [c9]past-due payments greater than three months should only be more risky, even if the model is unsure of the specific predictions it makes. Using this idea, we generalize the mean-variance optimization problem proposed by Chen (2022). This results in a complex **non-convex mixed-integer nonlinear programming** problem. Generally, such a problem is difficult to solve, as discussed in Burer & Letchford (2012). Existing methods may not be capable of solving this problem efficiently. As an example, Chen (2022) does not consider the discrete nature of features and non-convexity. By taking advantage of the monotonic property of models, we propose to use the **fast marching method (FMM)** to find the **global** [c10]optimizer to the problem. The FMM algorithm (Tsitsiklis, 1995; Sethian, 1996; Helmsen et al., 1996) was originally proposed for tracing interface evolution by solving partial differential equations and has been very successful. By utilizing general types of monotonicity, we extend the FMM to solve our optimization problem. By using empirical examples, we demonstrate that our method has the capability of providing reliable bounds to unconfident predictions and enhances the prediction of uncertainty.

---

[c5] *Text added.*
[c6] ~~past dues~~
[c7] ~~past dues~~
[c8] ~~past dues~~
[c1] ~~past due~~
[c2] ~~then the model should predict that the probability of default is increasing, otherwise the model would be unfair.~~
[c3] ~~past due~~
[c4] ~~past dues~~
[c5] ~~past dues~~
[c6] ~~past due~~
[c7] ~~past due~~
[c8] ~~past dues~~
[c9] ~~past dues~~
[c10] ~~solution~~

In this work, we make three major contributions:

- We generalize the two-stage framework by Chen (2022) with only individual monotonicity to general types of monotonicity. [c1]By incorporating pairwise monotonicity, it will be possible to search a larger search space for optimization, resulting in better global optimizers. [c2]

- The monotonicity-induced optimization geometry of the domain is studied, providing an intuitive understanding of the geometry and permitting the implementation of algorithms in practice.

- A fast marching method based on monotonicity is proposed to find the **global** [c3]optimizer to the complex **non-convex mixed-integer nonlinear programming** optimization. [c4]Empirical results indicate that the fast marching method with all monotonicity has improved the accuracy of the baseline method.

## 2 Prerequisites

For problem setup, assume we have $n$ samples [c5]$\{\mathbf{x}_i\}_{i=1}^n$ and $m$ features [c6]such that $\mathbf{x}_i \in \mathbb{R}^m$, the data-generating process is

$$y_i = f(\mathbf{x}_i) + \epsilon_i \tag{1}$$

for regression problems [c7], where $\mathbf{x}_i$ is a random input, $y_i$ is a label, $f$ is the ML model, and $\epsilon_i$ is the random noise, and

$$y_i|\mathbf{x}_i = \text{Bernoulli}(g^{-1}(f(\mathbf{x}_i))) \tag{2}$$

for binary classification problems, where $g$ is the link function (e.g., logistic function). For simplicity, we assume $x_j \in \mathbb{R}^+ \cup \{0\}$ [c8]where $x_j$ denotes the $j$th feature of $\mathbf{x}$. Assumptions of this type are common in cost-sharing problems (Friedman & Moulin, 1999) and are often reasonable for high-stakes applications. Then ML methods are applied to approximate $f$.

### 2.1 Individual and Pairwise Monotonicity

Monotonicity is crucial for ensuring conceptual soundness and fairness and is therefore often strictly required in high-stakes applications (Chen & Ye, 2023; Gupta et al., 2020; Liu et al., 2020). Throughout the paper, without loss of generality, we focus on the [c9]monotonically increasing functions. Suppose $f$ is individual monotonic with respect to $x_\alpha$ and $\neg\boldsymbol{\alpha}$ is the complement of $\alpha$, then the input $\mathbf{x}$ can be partitioned into $\mathbf{x} = (x_\alpha, \mathbf{x}_{\neg\boldsymbol{\alpha}})$. The well-known **individual monotonicity** is then defined as below.

**Definition 2.1.** *We say $f$ is **individually monotonic** with respect to $x_\alpha$ if $\forall x_\alpha, x'_\alpha, \mathbf{x}_{\neg\boldsymbol{\alpha}}$,*

$$f(x_\alpha, \mathbf{x}_{\neg\boldsymbol{\alpha}}) \leq f(x'_\alpha, \mathbf{x}_{\neg\boldsymbol{\alpha}}), x_\alpha \leq x'_\alpha. \tag{3}$$

[c10]Individual monotonicity is a common phenomenon in practice. In credit scoring, for instance, the probability of default should be individually monotonic with respect to the number of past dues. [c11]In criminology,

---

[c1] ~~In this way, a larger search space of optimization can be used for the solution, and thus tighter bounds can be provided.~~

[c2] ~~Consequently, pairwise monotonicity improves the results.~~

[c3] ~~solution~~

[c4] *Text added.*

[c5] *Text added.*

[c6] *Text added.*

[c7] *Text added.*

[c8] ~~for all $i$~~

[c9] ~~monotonic~~

[c10] *Text added.*

[c11] *Text added.*

the likelihood of recidivism is individually monotonic with respect to the number of past criminal charges. In education, the probability of admission should be monotonic in relation to a student's grade point average.

In practice, certain features are intrinsically more important than others, and **pairwise monotonicity** (Gupta et al., 2020; Cotter et al., 2019; Chen & Ye, 2023; 2022) describes these phenomena. Analog to equation 3, we partition $\mathbf{x} = (x_\beta, x_\gamma, \mathbf{x}_\neg)$. Without loss of generality, we assume $x_\beta$ is more important than $x_\gamma$. In addition, we require all features with pairwise monotonicity also satisfy individual monotonicity. There are two types of pairwise monotonicity. We start with the **strong pairwise monotonicity**.

**Definition 2.2.** *We say $f$ is **strongly monotonic** with respect to $x_\beta$ over $x_\gamma$ if $\forall x_\beta, x_\gamma, \forall \mathbf{x}_\neg, \forall c \in \mathbb{R}^+$*

$$f(x_\beta, x_\gamma + c, \mathbf{x}_\neg) \le f(x_\beta + c, x_\gamma, \mathbf{x}_\neg). \tag{4}$$

As an example, in criminology, for each additional crime, a felony is considered more serious than a misdemeanor for predicting the probability of recidivism. Next, we introduce the **weak pairwise monotonicity**.

**Definition 2.3.** *We say $f$ is **weakly monotonic** with respect to $x_\beta$ over $x_\gamma$ if $\forall x_\beta, x_\gamma$ s.t. $x_\beta = x_\gamma, \forall \mathbf{x}_\neg, \forall c \in \mathbb{R}^+$,*

$$f(x_\beta, x_\gamma + c, \mathbf{x}_\neg) \le f(x_\beta + c, x_\gamma, \mathbf{x}_\neg). \tag{5}$$

When it comes to predicting admission acceptance for STEM majors, math scores on the GRE are more important than verbal scores. In contrast to strong pairwise monotonicity, such comparisons are not always valid. If a student already has a good math score but a very poor verbal score, then an additional increase in verbal scores might increase more chances than math since universities want to ensure the student is capable of communicating effectively. The condition $x_\beta = x_\gamma$ ensures that the comparison is made at the same magnitude since the comparison is not always valid. The result is that, out of 170 points for math and verbal, a student with math 165 and verbal 150 has a greater chance of admission for STEM majors than a student with math 150 and verbal 165, but not necessarily a greater chance than a student with math 160 and verbal 155.

## 2.2 Detect Out-of-Distribution Data

[c1] Ovadia et al. (2019) [c2]presents a large-scale evaluation of different methods for quantifying predictive uncertainty across a variety of data modalities and architectures. Overall, they found that the ensemble methods by Lakshminarayanan et al. (2017) [c3]performed the best across most metrics and were the most robust to data shifts. Following Lakshminarayanan et al. (2017), models are trained $M$ times with random initialization and data shuffles in the entire dataset with [c4] $\{\widehat{f}(\mathbf{x}; \boldsymbol{\theta}_i)\}$, where $\{\boldsymbol{\theta}_i\}_{i=1}^M$ denote the parameters of each neural network in the ensemble. [c5]For the prediction, the average of the ensembles is used,

$$\widehat{\mu}(\mathbf{x}) = \frac{1}{M} \sum_{i=1}^M \widehat{f}(\mathbf{x}; \boldsymbol{\theta}_i). \tag{6}$$

The variance is used as a proxy for the level of model uncertainty and is calculated by

$$\widehat{\sigma}^2(\mathbf{x}) = \frac{1}{M-1} \sum_{i=1}^M (\widehat{f}(\mathbf{x}; \boldsymbol{\theta}_i) - \widehat{\mu}(\mathbf{x}))^2. \tag{7}$$

For a data point $\mathbf{x}$, when the variance is large, say $\widehat{\sigma}^2(\mathbf{x}) > \epsilon$ for a threshold $\epsilon$, it is considered too risky to make the prediction. Accordingly, the dataset is divided into confident and unconfident sets.

---

[c1] Lakshminarayanan et al. (2017) describes a simple yet effective approach to detecting OOD data and has shown to be the best performer by Ovadia et al. (2019). Ensemble methods are used.

[c2] *Text added.*

[c3] *Text added.*

[c4] $\{\widehat{f}_i(\mathbf{x}; \boldsymbol{\theta}_i)\}$

[c5] Models are then used as predictions based on their average,

# 3   Two-stage Method

We generalize the two-stage framework presented by Chen (2022) for general monotonicity. As a first step, using ensemble methods, we identify the unconfident set for OOD data. In the second stage, we solve a mean-variance optimization problem to provide bounds for the points in the unconfident set. Using general monotonicity, the search domain can be enlarged. [c6]This leads to a better optimizer, however, the problem becomes more complex with a larger domain, and more importantly, the geometry becomes more complex, possibly non-convex. Lastly, if bounds do not meet expectations, we leave them to human judgment.

## 3.1   Detect Out-of-Distribution Data

To detect the unconfident set, we wish to utilize the ensemble method. We demonstrate in the following Theorem that monotonicity is preserved by ensembles. [c1]In this regard, we can choose to use the ensemble $\widehat{\mu}(\mathbf{x})$ as the prediction.

**Proposition 3.1.** *If [c2]$\widehat{f}(\mathbf{x}; \boldsymbol{\theta}_i)$ achieves all monotonicity for all i, then $\widehat{\mu}$ preserves all monotonicity.*

The proof is left in Appendix A. We then apply the ensemble method and consider

$$\mathbb{S} = \{\mathbf{x} | \widehat{\sigma}^2(\mathbf{x}) \geq \epsilon\} \tag{8}$$

as the **unconfident set**. Similarly, we define $\mathbb{Q} = \{\mathbf{x} | \widehat{\sigma}^2(\mathbf{x}) < \epsilon\}$ as the confident set. [c3]A vertical bar | used in the set-builder notation throughout the manuscript denotes its meaning as "such that". In this example, $\mathbb{S}$ is the set of all samples $\mathbf{x}$ such that $\widehat{\sigma}^2(\mathbf{x}) \geq \epsilon$.

## 3.2   Mean-Variance Optimization

[c4]Predictions with low confidence are generally be excluded from decision making (Ovadia et al., 2019; OCC, 2021). We wish to provide more information about the unconfident predictions once the unconfident set has been determined. [c5] For an unconfident prediction $\mathbf{x} \in \mathbb{S}$ such that $\widehat{\sigma}^2(\mathbf{x}) \geq \epsilon$, it would be helpful to provide confident bounds in order to provide more information. Specifically, we wish to find $\mathbf{x}'$ and $\mathbf{x}''$ such that

$$\widehat{\mu}(\mathbf{x}) \in [\widehat{\mu}(\mathbf{x}'), \widehat{\mu}(\mathbf{x}'')], \text{ with } \widehat{\sigma}^2(\mathbf{x}') < \epsilon, \widehat{\sigma}^2(\mathbf{x}'') < \epsilon. \tag{9}$$

[c6] However, since we are unconfident about the prediction $\widehat{\mu}(\mathbf{x})$, we cannot directly compare $\widehat{\mu}(\mathbf{x})$ with $\widehat{\mu}(\mathbf{x}')$ and $\widehat{\mu}(\mathbf{x}'')$ for general models. For models that have been assumed to have monotonicity from domain knowledge, it is possible to draw comparisons directly from the monotonicity assumption. For example, in credit scoring, the probability of default should be monotonically increasing with respect to the number of past-due payments to penalize late payments. Thus, we know that an applicant with four past-due payments will have a greater default probability than an applicant with three past-due payments, assuming all other conditions are equal. Similarly, in criminology, the probability of recidivism should be monotonically increasing with respect to the number of past criminal changes; in college admission, a higher GPA should result in a higher chance of acceptance. This motivates us to define the following space so that monotonicity can be used to compare the values of function directly.

**Definition 3.2.** *We define the space $\Omega(\mathbf{x})$ as*

$$\Omega(\mathbf{x}) = \{\mathbf{x}' | f(\mathbf{x}') \underset{M}{\leq} f(\mathbf{x})\}. \tag{10}$$

---

[c6] ~~resulting in tighter bounds, but also complicating the problem.~~
[c1] *Text added.*
[c2] $\widehat{f_i}$
[c3] *Text added.*
[c4] *Text added.*
[c5] *Text added.*
[c6] *Text added.*

whereas $\underset{M}{\leq}$ denotes the **inequality by monotonicity**. *That is, we know $f(\mathbf{x}') \leq f(\mathbf{x})$ by the* [c7] *assumed monotonicity from the Definition 2.1, 2.2, 2.3* [c8]. [c9] *Similarly, $\underset{M}{\geq}$ can be defined.*

**Example 3.3.** [c1] *Here is a demonstration example from the credit scoring. Assume $f$ is the probability of default and $x_1, x_2$ is the number of past-due payments greater than three months and less than three months. To ensure fairness, we require that $f$ increase monotonically both with respect to $x_1$ and $x_2$ since past due behaviors should be penalized. Furthermore, we require that $f$ is monotonic over $x_1$ over $x_2$, since a past-due payment that exceeds three months should receive stronger punishment. As a result, $f$ must satisfy those monotonicity conditions. In accordance with OCC's guidelines, such monotonicity is essential for finance applications to meet conceptual soundness and fairness requirements.*

*For simplicity, let's focus only on the domain $0 \leq x_1, x_2 \leq 3$. Suppose now we have trained several neural networks with mean $\widehat{\mu}$ and variance $\widehat{\sigma}^2$. Assume that we are interested in $\mathbf{x} = (2, 0)$ with $\widehat{\sigma}^2(\mathbf{x}) \geq \epsilon$, therefore $\widehat{\mu}(\mathbf{x})$ could be unreliable. We know that $f(0, 0), f(1, 0) \leq f(2, 0)$ based on the individual monotonicity; we further know $f(0, 2), f(1, 1), f(0, 1) \leq f(2, 0)$ based on combining individual and pairwise monotonicity. This gives us $\Omega(\mathbf{x}) = \{(0, 0), (1, 0), (0, 2), (1, 1), (0, 1)\}$ such that $f(\mathbf{x}') \underset{M}{\leq} f(\mathbf{x})$ for $\mathbf{x}' \in \Omega(\mathbf{x})$. Although we don't know exactly what $f$ exactly is, we can determine $\Omega(\mathbf{x})$ solely based on monotonicity. We emphasize the use of $\underset{M}{\leq}$ on $f$ rather than the outputs from the function $\widehat{\mu}$. For example, $\widehat{\mu}$ may yield that $\widehat{\mu}(1, 2) \leq \widehat{\mu}(2, 0)$. However, we may find that this inequality is not reliable if we are uncertain about the $\widehat{\mu}(2, 0)$.*

[c2]

[c3]We wish to provide confident bounds $[\widehat{\mu}(\mathbf{x}'), \widehat{\mu}(\mathbf{x}'')]$ as tight as possible to provide more information. **Without loss of generality, we focus on finding the lower bound $\widehat{\mu}(\mathbf{x}')$, but finding the upper bound $\widehat{\mu}(\mathbf{x}'')$ is similar.** Specifically, we wish to maximize $\widehat{\mu}(\mathbf{x}')$ with $\widehat{\sigma}^2(\mathbf{x}') < \epsilon$, similar to the problems in the modern portfolio theory by Markowitz (1952). [c4] In summary, for each $\mathbf{x} \in \mathbb{S}$, we wish to solve the following optimization problem

$$\begin{cases} \max_{\mathbf{x}' \in \Omega(\mathbf{x})} \widehat{\mu}(\mathbf{x}'), \\ \text{subject to } \widehat{\sigma}^2(\mathbf{x}') < \epsilon, \\ \Omega(\mathbf{x}) = \{\mathbf{x}' | f(\mathbf{x}') \underset{M}{\leq} f(\mathbf{x})\}. \end{cases} \tag{11}$$

**Example 3.4.** [c5] *Following Example 3.3.* [c6]*Now suppose we have trained neural networks with mean prediction $\widehat{\mu}(\mathbf{x})$ and variance $\widehat{\sigma}^2(\mathbf{x})$ in Table 1.* [c7] *We consider the threshold $\epsilon = 10^{-3}$. Among all $\mathbf{x}' \in \Omega(\mathbf{x})$, we find for $\mathbf{x}' = (1, 1)$, $\widehat{\mu}(\mathbf{x}') = 0.4$ is the largest with the satisfied confidence criteria. Therefore, it serves as a confident lower bound for $\widehat{\mu}(\mathbf{x})$.*

[c8] *In a similar manner, we could find $\mathbf{x}''$ such that $f(\mathbf{x}) \underset{M}{\leq} f(\mathbf{x}'')$ with and $\widehat{\sigma}^2(\mathbf{x}'') < \epsilon$. Then $\widehat{\mu}(\mathbf{x}'')$ serves as a confident upper bound for $\widehat{\mu}(\mathbf{x})$. Overall, we are confident that $\widehat{\mu}(\mathbf{x}) \in [\widehat{\mu}(\mathbf{x}'), \widehat{\mu}(\mathbf{x}'')] = [0.4, 0.8]$.*

---

[c7] ~~monotonicity from domain knowledge~~

[c8] ~~(not from the function outputs)~~

[c9] *Text added.*

[c1] *Text added.*

[c2] **~~Without loss of generality, we focus on finding lower bounds, but finding upper bounds is similar.~~** ~~Suppose we are unconfident about a prediction at $\mathbf{x} \in \mathbb{S}$, we wish to find a confident prediction at $\mathbf{x}'$ with $f(\mathbf{x}') \leq f(\mathbf{x})$ known by monotonicity. Then we have a confident lower bound for $\widehat{\mu}(\mathbf{x})$.~~

[c3] *Text added.*

[c4] ~~Therefore, we wish to maximize $\widehat{\mu}(\mathbf{x}')$ and minimize $\widehat{\sigma}^2(\mathbf{x}')$. However, we cannot perform both optimizations simultaneously, similar to the problems in modern portfolio theory by Markowitz (1952). Hence, we consider only confident prediction with $\widehat{\sigma}^2(\mathbf{x}') < \epsilon$ and look for the largest lower bound $\widehat{\mu}(\mathbf{x}')$. For optimization, we should focus on the domain that can be determined by monotonicity, and the following definition is provided.~~

[c5] *Text added.*

[c6] *Text added.*

[c7] *Text added.*

[c8] *Text added.*

Table 1: [c11]An example of mean-variance optimization of equation 11. [c12]The threshold is $\epsilon = 10^{-3}$. We are interested in the prediction of $\mathbf{x} = (2,0)$, which is marked bold. By monotonicity, we marked points with $f(\mathbf{x}') \underset{M}{\leq} f(\mathbf{x})$ as red and $f(\mathbf{x}'') \underset{M}{\geq} f(\mathbf{x})$ as blue. The confident bounds are given by $[0.4, 0.8]$.

| | (0,0) | (0,1) | (1,0) | (0,2) | (1,1) | **(2,0)** | (0,3) | (1,2) | (2,1) | (3,0) |
|---|---|---|---|---|---|---|---|---|---|---|
| $\widehat{\mu}$ | 0 | 0.1 | 0.2 | 0.3 | 0.4 | 0.5 | 0.6 | 0.7 | 0.8 | 0.9 |
| $\widehat{\sigma}^2$ | $10^{-5}$ | $10^{-5}$ | $10^{-5}$ | $10^{-5}$ | $10^{-5}$ | $10^{-2}$ | $10^{-5}$ | $10^{-5}$ | $10^{-5}$ | $10^{-5}$ |

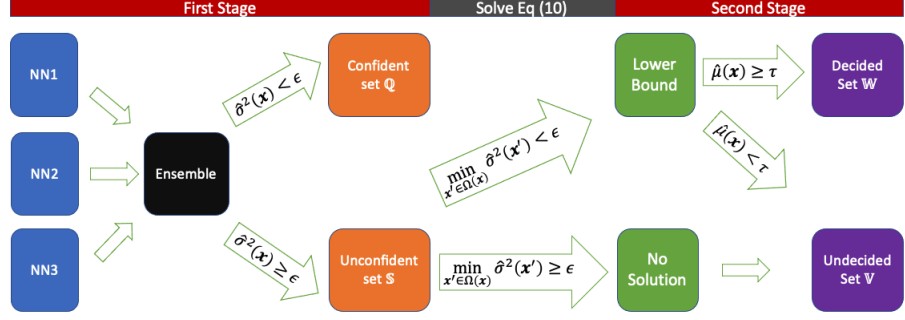

Figure 1: Two-stage framework

Although we can obtain the maximum value of $\widehat{\mu}$, it may not be useful if $\widehat{\mu}$ is too small. In practice, we would focus only on $\widehat{\mu} \geq \tau$, for some $\tau$ determined by users based on their risk appetites. We may be unable to find satisfactory lower bounds, in which case we leave the decision to human judgment. This may be the case, for example, if we have outliers for nonmonotonic features for which we lack domain expertise. Furthermore, if the variance of all points in the domain is high, there may be no solution. These data points are considered to be part of the **undecided set** $\mathbb{V}$ such that

$$\mathbb{V}(\tau) = \{\mathbf{x} | \mathbf{x} \in \mathbb{S} \text{ and } \min_{\mathbf{x}' \in \Omega(\mathbf{x})} \widehat{\sigma}^2(\mathbf{x}') > \epsilon\} \cup \{\mathbf{x} | \mathbf{x} \in \mathbb{S} \text{ and } \mathbf{x} \text{ solves equation 11 and } \widehat{\mu}(\mathbf{x}) < \tau\}. \tag{12}$$

Similarly, we define the decided set as $\mathbb{W}(\tau) = \{\mathbf{x} | \mathbf{x} \in \mathbb{S} \text{ and } \mathbf{x} \text{ solves equation 11 and } \widehat{\mu}(\mathbf{x}) \geq \tau\}$. A demonstration of the two-stage framework is provided in Figure 1.

## 4 Geometry of the Domain

We would like to provide a more explicit form for $\Omega(\mathbf{x})$. In the case of only individual monotonicity, $\Omega(\mathbf{x})$ is easily determined as a high-dimensional box. The presence of pairwise monotonicity allows us to have a larger search geometry $\Omega(\mathbf{x})$. However, $\Omega(\mathbf{x})$ is also much more complicated. With this study, we are able to obtain a better understanding of the geometry and also permit us to implement the algorithm in practice.

In the rest of the paper, we will ignore nonmonotonic features for the sake of simplicity, unless otherwise stated, since we cannot draw any conclusions from them. The features are divided into individual monotonic, weak pairwise monotonic, and strong pairwise monotonic parts, as $\mathbf{x} = (\mathbf{x}_S, \mathbf{x}_U, \mathbf{x}_P)$. For features with weak pairwise monotonicity in $U$, we give two lists $\mathbf{u}$ and $\mathbf{v}$ with $|\mathbf{u}| = |\mathbf{v}|$ such that $f$ is weakly pairwise monotonic with respect to $x_{u_i}$ over $x_{v_i}$ for $i = 1, \ldots, |\mathbf{u}|$. For strong pairwise monotonicity, we assume that there is a list $\mathbf{p}$ such that $f$ is strongly pairwise monotonic to $x_{p_j}$ over $x_{p_{j+1}}$ for $j = 1, \ldots, |\mathbf{p}| - 1$. All monotonic features follow individual monotonicity. This structure is sufficient for most applications, but more complicated structures can be considered if necessary. All proofs are left in Appendix A.

[c1]The following sections analyze the geometry induced by individual monotonicity, weak pairwise monotonicity, and strong pairwise monotonicity in Section 4.1, 4.2, and 4.3. [c2]Afterward, we will provide the results when all monotonicity has been considered in Section 4.4.

## 4.1 Individual Monotonicity

In the case that $x_1, \ldots, x_m$ only exhibit individual monotonicity, [c1]the geometry resulting from the individual monotonicity is a box.

**Proposition 4.1.** *Suppose $f$ is individually monotonic with respect to $x_1, \ldots, x_m$, then*

$$\Omega(\mathbf{x}) = \{\mathbf{x}'|\mathbf{x}' \le \mathbf{x}\} = \{(x_1', \ldots, x_m')|x_1' \le x_1, \ldots, x_m' \le x_m\}.$$

Figure 2a with $\mathbf{x} = (3, 1)$ is provided for demonstration. A reduction operator is defined for detecting data points that can be determined by individual monotonicity, which will be used later in the manuscript.

**Definition 4.2.** *We denote $\psi(\mathbf{x}, \mathbf{c})$ to reduce values of $\mathbf{x}$ by $\mathbf{c}$, that is,*

$$\psi(\mathbf{x}, \mathbf{c}) = \mathbf{x} - \mathbf{c}. \tag{13}$$

It is easy to check that if $\mathbf{x}' \in \Omega(\mathbf{x})$, then $\exists \mathbf{c}$ such that $\mathbf{x}' = \psi(\mathbf{x}, \mathbf{c})$.

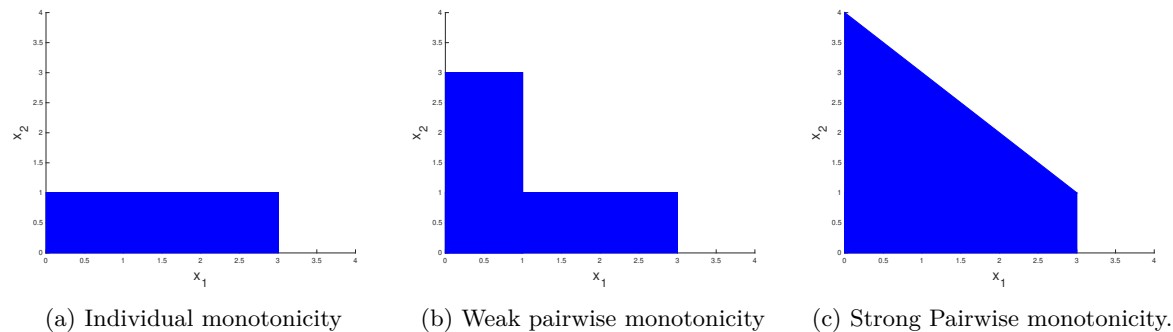

(a) Individual monotonicity     (b) Weak pairwise monotonicity     (c) Strong Pairwise monotonicity.

Figure 2: Geometry for $\Omega(\mathbf{x})$, where $\mathbf{x} = (x_1, x_2) = (3, 1)$. With the progression from individual monotonicity to weak pairwise monotonicity to strong pairwise monotonicity, we are able to obtain a larger geometry. In the case of weak pairwise monotonicity, the geometry may not be convex.

## 4.2 Weak Pairwise Monotonicity

Pairwise monotonicity presents a more challenging situation. Firstly, we will identify the maximum boundary points, followed by determining the interior points. [c2]By identifying the maximum boundary points, we will be able to determine whether a point belongs to the desired geometry.

A swap operator is defined in order to detect data points that can be identified by weak pairwise monotonicity.

**Definition 4.3.** *For indices $\beta$ and $\gamma$, we denote $\Gamma(\mathbf{x}, \beta, \gamma)$ to swap values of $x_\beta$ and $x_\gamma$ in $\mathbf{x}$, that is, if $\mathbf{x}' = \Gamma(\mathbf{x}, \beta, \gamma)$, then*

$$x_i' = \begin{cases} x_i & \text{if } i \ne \beta \text{ and } i \ne \gamma, \\ x_\gamma, & \text{if } i = \beta, \\ x_\beta, & \text{if } i = \gamma. \end{cases} \tag{14}$$

---

[c1] *Text added.*
[c2] *Text added.*
[c1] ~~we would have a box, as shown below.~~
[c2] *Text added.*

With the size of $x_\beta + x_\gamma$ fixed, we obtain the following proposition [c3]to identify whether we are able to make the comparison under weak pairwise monotonicity.

**Proposition 4.4.** *Suppose $f$ is weakly pairwise monotonic with respect to $x_\beta$ over $x_\gamma$, then*

$$f(\Gamma(\mathbf{x}, \beta, \gamma)) \underset{M}{\leq} f(\mathbf{x}), \; if \; x_\beta > x_\gamma. \tag{15}$$

As a result, we can identify the maximum boundary points of the domain [c1] [c2]Basically, maximum boundary points provide us with points at the boundary of the domain with the largest magnitudes. We provide the definition below.

**Definition 4.5.** *The set of maximum boundary points of $\Omega$ is defined as follows:*

$$\partial\Omega(\mathbf{x}) = \left\{ \mathbf{x}' \middle| \sum_{i=1}^m x_i' = \sum_{i=1}^m x_i, f(\mathbf{x}') \underset{M}{\leq} f(\mathbf{x}) \right\}. \tag{16}$$

[c3]As a next step, we would like to determine the maximum boundary points under weak pairwise monotonicity. We consider swapping by weak pairwise monotonicity to determine maximum boundary points. As a result, we provide the following proposition.

**Proposition 4.6.** *Suppose $f$ is weakly monotonic with respect to $x_{u_i}$ over $x_{v_i}$ for $i = 1, \ldots, |\mathbf{u}|$, then*

$$\partial\Omega(\mathbf{x}) = \bigcup_{i:x_{u_i} > x_{v_i}} \partial\Omega(\mathbf{x}'|\mathbf{x}' = \Gamma(\mathbf{x}, u_i, v_i)). \tag{17}$$

In other words, by fixing the size of $\sum_{i:i \in U} x_i$, we consider all possible swaps. Recursive definitions are used since swaps can be performed more than once. Clearly, for a new point $\mathbf{x}'$, if $\sum_{i=1}^m x_i' > \sum_{i=1}^m x_i$, we don't have enough information to compare it with $\mathbf{x}$. Conversely, we have the following theorem, [c4]indicates that all points in the domain are bounded by maximum boundary points.

**Theorem 4.7.** *Suppose $f$ is weakly monotonic with respect to $x_{u_i}$ over $x_{v_i}$ for $i = 1, \ldots, |\mathbf{u}|$, there exists a $\mathbf{x}'$ with $\sum_{i \in U} x_i' < \sum_{i \in U} x_i$ and $f(\mathbf{x}') \underset{M}{\leq} f(\mathbf{x})$, then there exists a $\widetilde{\mathbf{x}}' \in \partial\Omega(\mathbf{x})$ such that $\mathbf{x}' \leq \widetilde{\mathbf{x}}'$.*

By Theorem 4.7, we need only consider the "interior" of $\partial\Omega$ to determine the domain. It should be noted that in this case, the interior has a different definition.

**Definition 4.8.** *We say $\mathbf{x}$ is an interior point of $\partial\Omega$ if $\mathbf{x} \leq \mathbf{x}'$ for some $\mathbf{x}' \in \partial\Omega$. Correspondingly, we write $\mathbf{x} \leq \partial\Omega$ if $\mathbf{x} \leq \mathbf{x}'$ for some $\mathbf{x}'$ in $\partial\Omega$.*

We use this definition for better demonstration. If $|u| = |v| = 1$ and $x_u > x_v$, we have

$$\Omega(\mathbf{x}) = \{\mathbf{x}'|\mathbf{x}' \leq \mathbf{x}\} \cup \{\mathbf{x}'|\mathbf{x}' \leq \Gamma(\mathbf{x}, u, v)\}$$

More generally, we have the following proposition [c5]to provide a formula for the domain under weak pairwise monotonicity, as a result of Proposition 4.6 and Theorem 4.7.

**Proposition 4.9.** *Suppose $f$ is weakly monotonic with respect to $x_{u_i}$ over $x_{v_i}$, then*

$$\Omega(\mathbf{x}) = \{\mathbf{x}'|\mathbf{x}' \leq \mathbf{x}\} \cup \bigcup_{i:x_{u_i} > x_{v_i}} \Omega(\mathbf{x}'|\mathbf{x}' = \Gamma(\mathbf{x}, u_i, v_i)). \tag{18}$$

**Remark 4.10.** *The geometry becomes larger in the presence of weak pairwise monotonicity compared to only individual monotonicity as illustrated in Figure 2b. However, $\Omega(\mathbf{x})$ is not necessarily convex. Suppose, for example, that $f$ is weakly monotonic with respect to $x_1$ over $x_2$, and we have $\mathbf{x} = (3, 1)$, then we have*

$$\Omega(\mathbf{x}) = \{(x_1', x_2')|x_1' \leq 3, x_2' \leq 1\} \cup \{(x_1', x_2')|x_1' \leq 1, x_2' \leq 3\}.$$

*As shown in Figure 2b, it is evident that it is not convex, and thus complicate the optimization.*

---

[c3] ~~under weak pairwise monotonicity.~~
[c1] ~~, which we define as follows.~~
[c2] *Text added.*
[c3] *Text added.*
[c4] *Text added.*
[c5] *Text added.*

### 4.3 Strong Pairwise Monotonicity

We then consider the strong pairwise monotonicity. Suppose we have a list $\mathbf{p}$ such that $f$ is strongly pairwise monotonic with respect to $x_{p_i}$ over $x_{p_{i+1}}$ for $i = 1, \ldots, |\mathbf{p}| - 1$. As a starting point, we will consider the case in two dimensions. Assume that $f$ is strongly monotonic with respect to $x_1$ over $x_2$. Based on the strong pairwise monotonicity of $x_1 + x_2$, we can derive the following proposition.

**Proposition 4.11.** *Suppose $f$ is strongly pairwise monotonic with respect to $x_\beta$ over $x_\gamma$, if $x'_\beta \leq x_\beta, x'_\gamma = x_\beta + x_\gamma - x'_\beta$, then*

$$f(x'_\beta, x'_\gamma, \mathbf{x}_\neg) \underset{M}{\leq} f(x_\beta, x_\gamma, \mathbf{x}_\neg). \tag{19}$$

As a result, it can be shown that

$$\Omega(\mathbf{x}) = \{(x'_1, x'_2) | x'_1 \leq x_1, x'_2 \leq x_1 + x_2 - x'_1\}.$$

A simple example of $\mathbf{x} = (3, 1)$ is given in Figure 2c with a comparison to individual and weak pairwise monotonicity. Clearly, we would be able to have a larger searching geometry with strong pairwise monotonicity. [c1]In general, the following theorem applies to geometry induced by strong pairwise monotonicity.

**Theorem 4.12.** *For $f(x_1, \ldots, x_m)$, where $f$ is strongly pairwise monotonic with respect to $x_i$ over $x_{i+1}$ for $i = 1, \ldots, m - 1$, then*

$$\Omega(\mathbf{x}) = \left\{ \mathbf{x}' \middle| x'_i \leq \sum_{j=1}^{i} x_j - \sum_{j=1}^{i-1} x'_j, \forall i \right\}. \tag{20}$$

As a result of Theorem 4.12, [c2]we obtain the following proposition, which allows us to calculate the maximum boundary points.

**Proposition 4.13.** *For $f(x_1, \ldots, x_m)$, where $f$ is strongly pairwise monotonic with respect to $x_i$ over $x_{i+1}$ for $i = 1, \ldots, m - 1$, denote*

$$\varphi(\mathbf{x}, \mathbf{p}) = \left\{ \mathbf{x}' \middle| x'_i \leq \sum_{j=1}^{i} x_j - \sum_{j=1}^{i-1} x'_j, \forall i, \sum_{i=1}^{m} x_i = \sum_{i=1}^{m} x'_i \right\}, \tag{21}$$

*then*

$$\partial\Omega(\mathbf{x}) = \varphi(\mathbf{x}, \mathbf{p}). \tag{22}$$

*As a result, if there exists a $\mathbf{x}'$ with $\sum_{i=1}^{m} x'_i < \sum_{i=1}^{m} x_i$ and $f(\mathbf{x}') \underset{M}{\leq} f(\mathbf{x})$, then there exists a $\widetilde{\mathbf{x}}' \in \partial\Omega(\mathbf{x})$ such that $\mathbf{x}' \leq \widetilde{\mathbf{x}}'$.*

That is, $\mathbf{x}' \in \Omega(\mathbf{x})$ if and only if $\mathbf{x}'$ is interior of $\partial\Omega(\mathbf{x})$.

### 4.4 General Cases

In general, suppose all features are individual monotonic, $f$ is weakly monotonic with respect to $x_{u_i}$ over $x_{v_i}$ for lists $\mathbf{u}$ and $\mathbf{v}$ in $U$, and $f$ is strongly monotonic with respect to $p_j$ over $p_{j+1}$ for $j = 1, \ldots, |\mathbf{p}| - 1$. We generalize $\varphi$ by requiring that $\mathbf{x}'_\neg = \mathbf{x}_\neg$ for features not in $\mathbf{p}$ in equation 21. Then we write the geometry recursively as

$$\Omega(\mathbf{x}) = \{\mathbf{x}' | \mathbf{x}' \leq \mathbf{x}\} \cup \{\mathbf{x}' | \mathbf{x}' \leq \varphi(\mathbf{x}, \mathbf{p})\} \cup \bigcup_{i: x_{u_i} > x_{v_i}} \Omega(\mathbf{x}' | \mathbf{x}' = \Gamma(\mathbf{x}, u_i, v_i)). \tag{23}$$

---

[c1] ~~We consider the following theorem for more general cases.~~
[c2] ~~we have the following proposition.~~

## 5 Fast Marching Method

We discuss how to solve [c3] equation 11 in this section. The optimization is challenging because of the **nonlinearity** of $\widehat{\mu}(\mathbf{x})$ and $\widehat{\sigma}(\mathbf{x})$, **discrete** and continuous features, and potential **non-convex** geometry (from both $\mathbf{x}' \in \Omega(\mathbf{x})$ and constraints $\widehat{\sigma}^2(\mathbf{x}') < \epsilon$). Therefore, standard optimization algorithms may not be sufficient and difficulties of such problems are discussed by Burer & Letchford (2012). Chen (2022) neglects the discrete nature of some features and potential non-convexity. Based on the monotonicity results studied in Section 4, we would like to pursue a different approach to find a **global** [c4]maximizers.

We begin by binning the features, with discussions in Appendix E. After binning, as a convenience, we assume that $x_j \in \mathbb{Z}^+ \cup \{0\}$, $j = 1, \ldots, m$.

We would like to make use of monotonicity. Specifically, we want to go through the monotonic sequence

$$\begin{cases} \mathbf{x}^1 \to \mathbf{x}^2 \to \ldots, \\ \text{where } \widehat{\mu}(\mathbf{x}^i) \underset{M}{\geq} \widehat{\mu}(\mathbf{x}^{i+1}). \end{cases} \tag{24}$$

and we stop when $\widehat{\sigma}^2(\mathbf{x}^i) < \epsilon$. By brute-force calculation, all possible points in the space must be calculated and sorted, which can be very expensive. Therefore, we intend to carry out this process iteratively.

For each point $\mathbf{x}$, [c1]we want to check its neighbor points in the domain $\Omega(\mathbf{x})$ that have not yet been examined. Define $\mathbf{e}_i$ as

$$(\mathbf{e}_i)_j = \begin{cases} 1, & \text{if } j = i, \\ 0, & \text{otherwise.} \end{cases}$$

Each time we iterate, we explore $\psi(\mathbf{x}, \mathbf{e}_i)$ for all $i$, that is, we consider decreasing one unit of the feature. Due to Theorem 4.7 and Proposition 4.13, we only need to include maximum boundary points by pairwise monotonicity in the initial set defined recursively as

$$l(\mathbf{x}) = \mathbf{x} \cup \varphi(\mathbf{x}, \mathbf{p}) \cup \bigcup_{i : x_{u_i} > x_{v_i}} l(\mathbf{x}' | \mathbf{x}' = \Gamma(\mathbf{x}, u_i, v_i)). \tag{25}$$

In each iteration, we define our search as follows:

$$\phi(\mathbf{x}) = \bigcup_{i : x_i > 0} \psi(\mathbf{x}, \mathbf{e}_i). \tag{26}$$

As a result, we develop the marching method [c2](MM).

---

**Algorithm 1** (Fast) Marching Method ((F)MM)

---

1: **Inputs**: $\mathbf{x}$, $\widehat{\mu}(\mathbf{x})$, $\widehat{\sigma}(\mathbf{x})$, and a set $l$ defined in equation 25
2: **Outputs**: Return the **global** solution to equation 11 if exists
3: **while** $\widehat{\sigma}^2(\mathbf{x}) \geq \epsilon$ and $|l| > 0$ **do**
4:      $l = l \cup \{\mathbf{x}' | \mathbf{x}' \in \phi(\mathbf{x}) \text{ and } \mathbf{x}' \text{ has not been visited}\}$
5:      Return $\mathbf{x}$ as the element corresponds to maximum $\widehat{\mu}(\mathbf{x}')$ in $l$ and remove it from $l$ (by Heap)
6: **end while**

---

The most expensive calculation in the marching algorithm is to determine the maximum value in the set $l$. A straightforward calculation costs $\mathcal{O}(|l|)$. The heap data structure can accelerate such calculations, as discussed by (Tsitsiklis, 1995; Sethian, 1996; Helmsen et al., 1996). This algorithm is known as the **Fast Marching Method (FMM)**, which has been proven to be a highly effective method for tracing interface evolution by solving partial differential equations. As there are more insertions than extract-max operations, we use the Fibonacci heap by Fredman & Tarjan (1987), which has a lower insertion cost than the binary heap. Different from the original FMM, we utilize general monotonicity from domain knowledge.

---

[c3] ~~the optimization problem~~
[c4] ~~solution~~
[c1] ~~we want to explore its nearby~~
[c2] *Text added.*

Table 2: Summary results by the FMM using all monotonicity [c3]and the baseline Ipopt using only individual monotonicity. In most cases, FMM has improved accuracy.

| | DATASETS | $\tau = 0.5$ | 0.4 | $\frac{|\mathbb{V}|}{|\mathbb{S}|}$ (%) 0.3 | 0.2 | 0.1 | 0 | MEAN-ITER |
|---|---|---|---|---|---|---|---|---|
| FMM | GMSC | 77.4 | 64.5 | 43.2 | 15.7 | 7.9 | 1.0 | **29** |
| | COMPAS | 97.7 | 86.9 | 80.1 | 74.0 | 72.5 | 72.2 | **4** |
| | Law | 78.9 | 58.3 | 39.0 | 21.1 | 14.2 | 14.2 | 343 |
| | Life-Science | 75.0 | 55.0 | 30.0 | 10.0 | 0.0 | 0.0 | **9** |
| | Mammography | 77.8 | 76.7 | 76.4 | 74.1 | 71.7 | 70.7 | **12** |
| Baseline | GMSC | 91.5 | 84.4 | 75.0 | 55.0 | 32.8 | 12.1 | 66 |
| | COMPAS | 98.9 | 91.6 | 87.3 | 81.7 | 79.7 | 79.4 | 36 |
| | Law | 80.0 | 65.3 | 50.9 | 42.0 | 34.0 | 33.4 | **50** |
| | Life-Science | 100.0 | 90.0 | 70.0 | 40.0 | 30.0 | 25.0 | 10 |
| | Mammography | 78.1 | 76.7 | 76.4 | 76.2 | 74.6 | 70.7 | 20 |

## 5.1 Analysis of the Algorithm

The algorithm is now briefly analyzed with the proof left in Appendix A. The following proposition shows that the algorithm marches monotonically.

**Proposition 5.1.** *MM searches for the solution in a monotonic non-increasing order of $\widehat{\mu}$.*

To ensure that we do not miss any points during the march, we provide the following proposition.

**Proposition 5.2.** *When MM runs to the end with $l = \{\}$, all points in the domain have been explored.*

Let us suppose that the iteration stops after $N$ steps. Then the space complexity is $\mathcal{O}(mN)$. It is estimated that the amortized time for inserting is $\Theta(1)$ and deleting the maximum key is $\Theta(\log(|l|))$. We further assume that the calculation of $\widehat{\mu}(\mathbf{x})$ and $\widehat{\sigma}(\mathbf{x})$ is $C$. As a result, at each iteration, the amortized time is $\Theta(Cm + \log(|l|))$. The overall cost of FMM is

$$\text{Cost}_{FMM} = \Theta((Cm + \log(m))N + N\log(N)).$$

As a comparison, the cost of MM is

$$\text{Cost}_{MM} = \mathcal{O}(CmN + mN^2).$$

As a result, the operation of finding the maximum values will be very expensive for large $N$.

**Remark 5.3.** *In practice, the most expensive part is $CmN$, where $C$ depends on the architectures of models.*

# 6 Empirical Example

We provide examples in finance, criminology, education, life science, and healthcare. The results are summarized in Table 2. In all experiments, the monotonic groves of neural additive models (MGNAMs) proposed by Chen & Ye (2023) are used. [c1]An overview of MGNAMs is provided in Appendix B. For all examples, we use ten models ($M = 10$) for ensembles and threshold $\epsilon = 10^{-3}$. Note models and thresholds are not unique choices and we provide more discussions in Appendix E. We provide detailed analyses for two datasets and leave the remaining examples to the Appendix C.

## 6.1 Finance - Credit Scoring

We use the Kaggle credit score dataset, Give Me Some Credit (GSMC). Without loss of generality, we let $x_1 - x_3$ denote the number of past dues and their duration: 90+ days, 60-89 days, and 30-59 days. By domain

---
[c1] *Text added.*

Table 3: The comparison for the optimization 11 by considering different monotonicity for the GMSC dataset with samples affected by pairwise monotonicity. Pairwise monotonicity improves the result.

| | INDIVIDUAL | | | WEAK | | | STRONG | |
|---|---|---|---|---|---|---|---|---|
| $\tau$ | $\frac{|\mathbb{U}|}{|\mathbb{T}|}(\%)$ | MEAN-ITER | $\tau$ | $\frac{|\mathbb{U}|}{|\mathbb{T}|}(\%)$ | MEAN-ITER | $\tau$ | $\frac{|\mathbb{U}|}{|\mathbb{T}|}(\%)$ | MEAN-ITER |
| 0.5 | 72.4 | 15 | 0.5 | 71.9 | 19 | 0.5 | 70.0 | 24 |
| 0.4 | 58.8 | 16 | 0.4 | 57.7 | 19 | 0.4 | 54.6 | 24 |
| 0.3 | 35.5 | 16 | 0.3 | 33.7 | 19 | 0.3 | 32.3 | 24 |
| 0.2 | 11.8 | 18 | 0.2 | 9.0 | 22 | 0.2 | 8.8 | 28 |
| 0.1 | 6.0 | 19 | 0.1 | 4.6 | 24 | 0.1 | 4.6 | 31 |
| 0 | 3.6 | 21 | 0 | 3.6 | 27 | 0 | 3.6 | 35 |

knowledge, the probability of default is strongly monotonic with respect to $x_1$ over $x_2$ over $x_3$. Furthermore, we impose individual monotonicity for monthly income ($x_4$) and number of dependents ($x_5$).

### 6.1.1 Stage I - Detect OOD Data

As the first step, we apply the ensemble method to detect OOD data. Running the experiment with the entire dataset leads to the identification of approximately 2.8% of the data as uncertain samples, which are therefore categorized in the **unconfident set** $\mathbb{S}$. A common example would be an applicant with a high amount of past dues, which is very rare in the dataset. Considering the rarity of this prediction, it makes sense that our model is unconfident about it. The existence of OOD data seems to be commonplace. Consequently, we should exercise caution when applying our models to new situations.

### 6.1.2 Stage II - Finding Lower Bounds

Next, we use the FMM to solve the mean-variance optimization problem 11. Predictions higher than $\tau$ are left in the **undecided set** $\mathbb{V}$. Our analysis takes into account a variety of choices of $\tau$, which can be selected by users according to their risk appetites. It is important to emphasize that while $\tau = 50\%$ is a natural choice for image classification applications, it is not necessarily the best choice for credit scoring applications. Credit scoring aims to accurately predict the probability of default, and 50% is already a very high probability. For accuracy, we calculate $\frac{|\mathbb{V}|}{|\mathbb{S}|} \in [0, 1]$, which is used to determine the ratio of unconfident samples that cannot be provided with reliable lower bounds. Thus, 0 suggests that confident lower bounds are provided for all unconfident samples, whereas 1 suggests that no confident lower bounds are provided.

**Overall results.** Our findings are summarized below, based on Table 2. As a result of considering lower $\tau$, we are able to determine more confident predictions, as expected. Based on strong pairwise monotonicity, only 0.22% of the entire dataset is undecided when $\tau = 0.1$. [c1]

**Comparison using different monotonicity.** Further comparisons are made by considering individual, weak pairwise, and strong pairwise monotonicity. It should be noted that in this example, $x_1 - x_3$ exhibits strong pairwise monotonicity, which implies weak pairwise monotonicity. [c2] We focus on samples that are affected by pairwise monotonicity and denote the set that includes these samples as $\mathbb{T}$. It is important to note that not all samples are affected by pairwise monotonicity; for example, $(x_1, x_2, x_3) = (0, 0, 3)$. In such a scenario, the FMM would produce the same result regardless of whether pairwise monotonicity is considered or not, and therefore it is not of interest to us. Similarly, we denote the undecided set out of $\mathbb{T}$ as $\mathbb{U}$. The results are documented in Table 3. Based on the table, it can be seen that pairwise monotonicity provides consistent improvements for different values of $\tau$, suggesting that pairwise monotonicity should be used. The number of iterations is not significantly increased by including pairwise monotonicity, demonstrating the effectiveness of FMM.

---

[c1] ~~FMM has a mean number of iterations of 29 for each optimization, demonstrating its efficacy. In this regard, our method has proven to be successful.~~

[c2] ~~The case of individual monotonicity is similar to the work done by Chen (2022), but the FMM accounts for discrete features and offers global solutions.~~

Table 4: A successful example of GMSC.

| ITER | $x_1$ | $x_2$ | $x_3$ | $x_4$ | $x_5$ | MEAN | VARIANCE |
|---|---|---|---|---|---|---|---|
| 0 | 3 | 1 | 3 | 1 | 2 | 0.6071 | 0.00165 |
| 1 | 3 | 0 | 4 | 1 | 2 | 0.6034 | 0.00159 |
| 2 | 3 | 1 | 2 | 1 | 2 | 0.5989 | 0.00151 |
| 3 | 3 | 0 | 3 | 1 | 2 | 0.5935 | 0.00142 |
| 4 | 3 | 1 | 3 | 1 | 1 | 0.5918 | 0.00174 |
| 5 | 2 | 2 | 3 | 1 | 2 | 0.5881 | 0.00103 |
| 6 | 3 | 0 | 4 | 1 | 1 | 0.5880 | 0.00168 |
| 7 | 3 | 1 | 1 | 1 | 2 | 0.5864 | 0.00135 |
| 8 | 3 | 1 | 2 | 1 | 1 | 0.5834 | 0.00159 |
| 9 | 2 | 1 | 4 | 1 | 2 | 0.5828 | 0.00103 |
| 10 | 2 | 2 | 2 | 1 | 2 | 0.5793 | 0.00092 |

**A successful example.** To better demonstrate how FMM performs, we provide a successful example with

$$\mathbf{x} = \begin{bmatrix} 3 & 1 & 3 & 1 & 2 & 0.96 & 0.38 & 9 & 1 & 40 \end{bmatrix}.$$

The iterations of FMM are recorded in Table 4. By following a monotonic sequence, variance has been reduced to meet the threshold. The reason for this result is that a large number of past dues are rare. The overall number of past dues $(x_1 + x_2 + x_3)$ is greater than 7 in only 232 samples. As the number of past dues decreases, the model becomes more confident in its prediction. Additionally, strong pairwise monotonicity is necessary in order to get from $(x_1, x_2, x_3) = (3, 1, 3)$ to $(x_1, x_2, x_3) = (2, 2, 2)$.

**An unsuccessful example.** Unfortunately, not all cases can result in positive outcomes. In some cases, it may not be possible to reduce the variance to the desired level. [c1]In other words, for equation 11, [c2]we might have

$$\min_{\mathbf{x}' \in \Omega(\mathbf{x})} \widehat{\sigma}^2(\mathbf{x}') \geq \epsilon.$$

[c3]The main reason for this is that the uncertainty is primarily derived from nonmonotonic features. More specifically, suppose we split $\mathbf{x}$ into monotonic and nonmonotonic parts as $\mathbf{x} = (\mathbf{x}_{\boldsymbol{\alpha}}, \mathbf{x}_{\neg})$, if $\mathbf{x}_{\neg}$ is the main reason that $\mathbf{x}$ is being OOD, then the maximize may not exist in equation 11.

Here is an example,

$$\mathbf{x} = \begin{bmatrix} 0 & 0 & 1 & 1 & 2 & 0.90 & 1.06 & 25 & 10 & 57 \end{bmatrix}. \tag{27}$$

The iterations of FMM are recorded in Table 5. In spite of the fact that FMM has been run to the end and the variance has been significantly reduced, [c4]we are still unable to find the global maximizer for equation 11. In this example, it appears that $x_9$ is a large value that is substantially different from the mean value of 1. In fact, there are only 113 samples with $x_9 \geq 10$, which suggests that this feature is quite rare and may result in a high level of variance.

[c9]In order to determine which feature has the greatest influence on unconfident predictions, it is possible to apply the attribution method to the feature importance. By preserving desired axioms, attribution methods allocate attributions in a fair manner. A popular method that has been very successful is Shapley value (Shapley et al., 1953; Lundberg & Lee, 2017). [c10] We use the baseline Shapley value (BShap) (Sundararajan & Najmi, 2020) [c11]to allocate attributions to the difference of $\widehat{\sigma}^2$ between $\mathbf{x}$ and a confident baseline point $\mathbf{x}'$. A detailed description of the BShap can be found in Appendix G.

---

[c1] *Text added.*
[c2] *Text added.*
[c3] *Text added.*
[c4] ~~we are still unable to come up with a satisfactory bound.~~
[c9] *Text added.*
[c10] *Text added.*
[c11] *Text added.*

Table 5: An unsuccessful example of GMSC.

| ITER | $x_1$ | $x_2$ | $x_3$ | $x_4$ | $x_5$ | MEAN | VARIANCE |
|------|-------|-------|-------|-------|-------|--------|----------|
| 0 | 0 | 0 | 1 | 1 | 2 | 0.4182 | 0.00375 |
| 1 | 0 | 0 | 1 | 1 | 1 | 0.4028 | 0.00365 |
| 2 | 0 | 0 | 1 | 1 | 0 | 0.3820 | 0.00328 |
| 3 | 0 | 0 | 1 | 0 | 2 | 0.3770 | 0.00431 |
| 4 | 0 | 0 | 1 | 0 | 1 | 0.3622 | 0.00414 |
| 5 | 0 | 0 | 1 | 0 | 0 | 0.3422 | 0.00363 |
| 6 | 0 | 0 | 0 | 1 | 2 | 0.2335 | 0.00195 |
| 7 | 0 | 0 | 0 | 1 | 1 | 0.2223 | 0.00181 |
| 8 | 0 | 0 | 0 | 1 | 0 | 0.2074 | 0.00154 |
| 9 | 0 | 0 | 0 | 0 | 2 | 0.2043 | 0.00186 |
| 10 | 0 | 0 | 0 | 0 | 1 | 0.1941 | 0.00171 |
| 11 | 0 | 0 | 0 | 0 | 0 | 0.1806 | 0.00142 |

Figure 3: [c7]BShap for $\widehat{\sigma}^2(\mathbf{x})$ of the unsuccessful example in equation 27. [c8]For better demonstration, we normalize the BShap values such that $\sum_{i=1}^{m} |\text{BS}_i| = 1$. $x_9$ is the dominant cause of high variance.

[c12]We apply BShap to $\mathbf{x}$ in equation 27 [c13] with the baseline $\mathbf{x}'$ as the average of all data. The result is shown in Figure 3. [c14]Based on our results, it appears that $x_9$ is the main cause of unconfident predictions. However, $x_9$ is not considered a monotonic feature, we cannot [c15]reduce variance by changing $x_9$. If additional domain knowledge is included, it may be possible to [c16]further reduce variance. For example, local monotonicity might be imposed instead of global monotonicity (the probability of default increases after $N$ loans). This result encourages us to incorporate more domain knowledge into the model.

## 6.2 Criminology - Recidivism

We present another example with a less satisfactory result and analyze the reason and potential remedy. In criminology, we examine the prediction of recidivism using the Correctional Offender Management Profiling for Alternative Sanctions (COMPAS) (Pro, 2016). There are four monotonic features. The overall result is recorded in Table 2 and the comparison of individual and pairwise monotonicity is in Table 6. Although we can obtain reliable bounds for some data, the performance is not as good as that of the GMSC dataset.

---

[c12] *Text added.*
[c13] *Text added.*
[c14] *Text added.*
[c15] provide further information
[c16] provide further information

**Reasons for the less satisfactory result.** There is a difficulty encountered when using the FMM to reduce the variance of OOD data. [c1]For example, when $\tau = 0.3$, there are only 20% points in the unconfident set that find their global maximizers. Age appears to be a significant factor, as demonstrated in Appendix C, but there is no indication that it is a global monotonic feature. The relationship between age and the prediction variance, however, shows a clear pattern, as in Figure 4. Roughly speaking, model predictions are much less confident for young people. We expect the performance to improve significantly if we can incorporate further domain knowledge regarding age, such as local monotonicity for example. Even so, because we are not experts in the field, we do not impose further limitations on age here to avoid unfair treatment.

Table 6: The comparison of the optimization 11 by considering different monotonicity for the COMPAS dataset with samples affected by pairwise monotonicity. Pairwise monotonicity improves the result.

| | INDIVIDUAL | | | STRONG | |
|---|---|---|---|---|---|
| $\tau$ | $\frac{|\mathbb{U}|}{|\mathbb{T}|}(\%)$ | MEAN-ITER | $\tau$ | $\frac{|\mathbb{U}|}{|\mathbb{T}|}(\%)$ | MEAN-ITER |
| 0.5 | 73.0 | 4 | 0.5 | 67.6 | 5 |
| 0.4 | 70.3 | 7 | 0.4 | 64.9 | 9 |
| 0.3 | 70.3 | 7 | 0.3 | 62.3 | 10 |
| 0.2 | 67.6 | 8 | 0.2 | 62.3 | 10 |
| 0.1 | 67.6 | 8 | 0.1 | 62.3 | 11 |
| 0 | 67.6 | 8 | 0 | 62.3 | 11 |

## 6.3 Comparison to the Existing Method

[c1]The results of the present study are compared with those of Chen (2022). In Chen (2022)'s work, [c2]only individual monotonicity is assumed. If $S$ contains the index of all monotonic features such that $\mathbf{x} = (\mathbf{x}_S, \mathbf{x}_\neg)$, then equation 11 reduces to

$$
\begin{cases}
\max_{\mathbf{x}' \in \Omega(\mathbf{x})} \widehat{\mu}(\mathbf{x}'), \\
\text{subject to } \widehat{\sigma}^2(\mathbf{x}') < \epsilon, \\
\Omega(\mathbf{x}) = \{\mathbf{x}' | \mathbf{x}'_S \leq \mathbf{x}_S\}.
\end{cases}
$$

[c3]Now that the geometry has been explicitly specified, we follow Chen (2022) [c4] and use the existing non-linear programming optimization package. In our experiments, the Interior Point Optimizer[c4] (Ipopt) (Hart et al., 2017; 2011) [c5]is used. More details are provided in Appendix D.

[c6] A summary of the results is presented in Table 2. [c7] It should be noted that smaller $\tau$ would result in more global maximizers. Moreover, our method is guaranteed to have better accuracy since it determines the global maximizer and pairwise monotonicity is included, whereas Ipopt only finds the local maximizer with individual monotonicity. The algorithm with fewer mean iterations is bolded. Overall, we have observed that our method has improved accuracy greatly, with the exception of the Mammography dataset. In the case of the Mammography dataset, we find that the model is generally not confident about its prediction (the percentage of OOD data is greater than 50%). Thus, finding good maxima is difficult. Secondly, iteration numbers are not necessarily worse than the baseline method, except for the law school dataset. On the law school dataset, we observe that it is quite difficult for the FMM for some data points to find the global optimizer and run the algorithm to the end, thus resulting in a high number of iterations.

---

[c1] ~~For example, there are only 20% of which, find the confident lower bound when $\tau = 0.3$.~~

[c1] *Text added.*

[c2] *Text added.*

[c3] *Text added.*

[c4] *Text added.*

[c4] https://coin-or.github.io/Ipopt/

[c5] *Text added.*

[c6] *Text added.*

[c7] *Text added.*

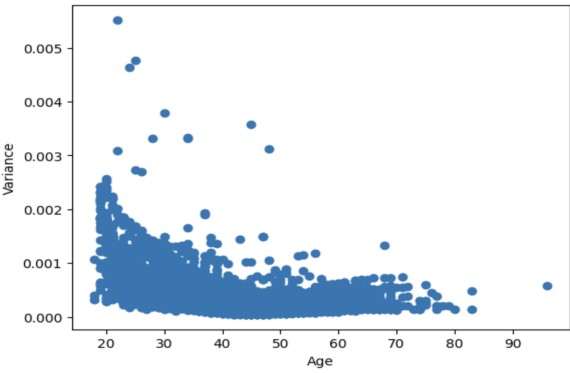

Figure 4: Variance vs Age in the COMPAS dataset.

## 7 Related Work

### 7.1 Relationship to Monotonic Models

[c8]There has been considerable interest in monotonic models in the ML community. In general, existing methods can be divided into two groups: (1) Monotonicity by constructions (Runje & Shankaranarayana, 2023; You et al., 2017; Milani Fard et al., 2016; Daniels & Velikova, 2010; Sill, 1997) and (2) Monotonicity by regularization (Liu et al., 2020; Bakst et al., 2020; Sivaraman et al., 2020; Sill & Abu-Mostafa, 1996). [c9]Both methods have been successful and have been applied to a wide range of applications.

[c1]Despite the many successes in monotonic models, these approaches mainly focus on individual monotonicity, and less attention is paid to pairwise monotonicity. A regularization approach is used by Gupta et al. (2020) [c2]to enforce individual and strong pairwise monotonicity for linear models, generalized additive models, and the nonlinear function class of multi-layer lattice models. For neural additive models, Chen & Ye (2022) [c3]enforces individual and weak pairwise monotonicity through regularization. Using a transparent architecture of MGNAMs, Chen & Ye (2023) [c4]enforces individual monotonicity, weak pairwise monotonicity, and strong pairwise monotonicity by regularization. We use MGNAMs here because they incorporate all the monotonicity we require.

[c5]Despite the fact that our methods are based on monotonic models as in equation 11, [c6]the focus of this research is not on how monotonic models are trained but rather on how monotonicity can be further utilized to reduce predictive uncertainty. In this way, our method can be applied to general monotonic models to reduce their predictive uncertainty.

### 7.2 Relationship to Out-of-distribution Detection

[c7]There have been studies of OOD detection methods that have been successful, including maximum softmax probability (Hendrycks & Gimpel, 2016), [c8]temperature scaling (Guo et al., 2017), [c9]Monte-Carlo dropout (Gal & Ghahramani, 2016; Srivastava et al., 2015), [c10]ensemble methods (Lakshminarayanan et al., 2017),

---

[c8] *Text added.*
[c9] *Text added.*
[c1] *Text added.*
[c2] *Text added.*
[c3] *Text added.*
[c4] *Text added.*
[c5] *Text added.*
[c6] *Text added.*
[c7] *Text added.*
[c8] *Text added.*
[c9] *Text added.*
[c10] *Text added.*

[c11]stochastic variational Bayesian inference (Blundell et al., 2015; Graves, 2011; Louizos & Welling, 2017), [c12]and approximated Bayesian inference based on the last layer (Riquelme et al., 2018). [c13]A detailed comparison of these methods can be found in Ovadia et al. (2019); Yang et al. (2022).

[c1]In traditional OOD tasks, the main focus is on detecting OOD data. As a result, the system could abstain from making decisions due to low confidence(Ovadia et al., 2019; OCC, 2021). [c2]We focus on the case of underlying models that have prior knowledge of monotonicity, and how a confident alternative point can be derived by assuming monotonicity to provide additional information of unconfident predictions. Therefore, the choice of OOD methods may not be unique. As part of this work, we examine ensemble methods and use their variance as a proxy for uncertainty. It is possible to extend this approach by considering other measures of uncertainty. A simple way to accomplish this is to switch $\widehat{\sigma}^2(\mathbf{x}) < \epsilon$ to other constraints in equation 11.

### 7.3 Relationship to Fast Marching Methods for Solving the Eikonal Equation

[c3]Originally, the FMM (Tsitsiklis, 1995; Sethian, 1996; Helmsen et al., 1996) [c4] was proposed as a solution to the Eikonal equation, which is a partial differential equation of the following form:

$$\begin{cases} |\nabla u(\mathbf{x})| & = \frac{1}{f(\mathbf{x})}, \text{ for } \mathbf{x} \in \Omega, \\ u(\mathbf{x}) & = 0, \text{ for } \mathbf{x} \in \partial\Omega, \end{cases} \tag{28}$$

[c5]where $|\cdot|$ is the Euclidean norm, $\nabla$ is the gradient, $f$ is given, $\Omega$ is the domain, and $\partial\Omega$ is the boundary of the domain. Such problems describe the evolution of a closed surface as a function of time $u$ with speed $f$ in the normal direction at a point $\mathbf{x}$ on the propagating surface. The speed function $f$ is specified, and the time $u$ at which the contour crosses a point $\mathbf{x}$ is calculated by solving the equation. Alternatively, $u(\mathbf{x})$ may be thought of as the minimum amount of time needed to reach $\partial\Omega$ from $\mathbf{x}$. [c6]As a result, the original Eikonal equation 28 [c7]differs completely from equation 11 [c8]we are attempting to solve.

Appendix F [c9]describes the original FMM. Based on the optimal control interpretation of the problem, the FMM constructs a solution outwards starting from the boundary values. In other words, the FMM calculates a sequence of monotonically non-decreasing solutions and utilizes the heap algorithm to accelerate the calculation of finding the minimum value and removing it from the set. Although the original problem of interest is completely different, we formulate our method in the form of a monotonic sequence as in equation 24. [c10]Using the monotonic properties of the model, we propose finding the global maximum of the equation by iteratively searching the points by equation 26 [c11]from the set $l$ in Algorithm 1. [c12]After this, the acceleration of finding the maximum value and removing it from the set is performed using the FMM, as was the case with the original technique to solve the Eikonal equation.

## 8 Limitations and Future Work

In this paper, we demonstrate how to exploit general monotonicity to reduce prediction uncertainty by providing a general optimization framework and a fast marching method. Overall, pairwise monotonicity has enhanced the performance and the fast marching method provides a global solution with reasonable

---

[c11] *Text added.*
[c12] *Text added.*
[c13] *Text added.*
[c1] *Text added.*
[c2] *Text added.*
[c3] *Text added.*
[c4] *Text added.*
[c5] *Text added.*
[c6] *Text added.*
[c7] *Text added.*
[c8] *Text added.*
[c9] *Text added.*
[c10] *Text added.*
[c11] *Text added.*
[c12] *Text added.*

iterations, making it a suitable benchmark for future research. There are some limitations of our current approach, as summarized below. Correspondingly, we propose future research.

1. The major limitation of the current FMM is that it does not consider the behavior of prediction variance, thus it may take a considerable number of iterations, especially for high-dimensional problems. We plan to utilize properties of model variances to improve the FMM's search process.

2. In general, FMM performs better when more features exhibit monotonicity, especially important features. It should be noted, however, that some important features may not exhibit monotonicity, at least not globally. The performance may be improved by applying other domain knowledge (Gupta et al., 2020; 2018) and imposing local monotonicity. Such a direction will be explored.

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

# A   MISSING PROOFS

This section contains detailed proofs of the results that are missing in the main paper.

## A.1   Proof of Theorem 3.1

*Proof.* Suppose $\widehat{f}_i$ is individually monotonic to $x_\alpha$, then

$$\widehat{f}_i(x_\alpha, \mathbf{x}_\neg) \underset{M}{\leq} \widehat{f}_i(x'_\alpha, \mathbf{x}_\neg), \text{ if } x_\alpha \leq x'_\alpha.$$

If this is true for all $i$, then for $x'_\alpha \geq x_\alpha$, we have

$$\frac{1}{M} \sum_{i=1}^{M} \widehat{f}_i(x_\alpha, \mathbf{x}_\neg) \underset{M}{\leq} \frac{1}{M} \sum_{i=1}^{M} \widehat{f}_i(x'_\alpha, \mathbf{x}_\neg)$$

Suppose $\widehat{f}_i$ is weakly pairwise monotonic with respect to $x_\beta$ over $x_\gamma$, then for $x_\beta = x_\gamma$, we have

$$\widehat{f}_i(x_\beta, x_\gamma + c, \mathbf{x}_\neg) \underset{M}{\leq} \widehat{f}_i(x_\beta + c, x_\gamma, \mathbf{x}_\neg), \forall c \in \mathbb{R}^+.$$

If this is true for all $i$, then we have

$$\frac{1}{M} \sum_{i=1}^{M} \widehat{f}_i(x_\beta, x_\gamma + c, \mathbf{x}_\neg) \underset{M}{\leq} \frac{1}{M} \sum_{i=1}^{M} \widehat{f}_i(x_\beta + c, x_\gamma, \mathbf{x}_\neg).$$

A similar conclusion can be drawn for strong pairwise monotonicity. $\qquad \square$

## A.2   Proof of Proposition 4.1

*Proof.* If $\mathbf{x}' \in \Omega(\mathbf{x})$, then $f(\mathbf{x}') \underset{M}{\leq} f(\mathbf{x})$ by definition. Conversely, if $x'_i > x_i$ for some $i$, we cannot draw any conclusions. $\qquad \square$

## A.3   Proof of Proposition 4.4

*Proof.* Without loss of generality, we write $\mathbf{x} = (x_\beta, x_\gamma, \mathbf{x}_\neg)$. Consider $\mathbf{x}' = (x_\gamma, x_\gamma, \mathbf{x}_\neg)$ and $c = x_\beta - x_\gamma > 0$, then by definition, we have

$$f(x_\gamma, x_\gamma + c, \mathbf{x}_\neg) \underset{M}{\leq} f(x_\gamma + c, x_\gamma, \mathbf{x}_\neg).$$

$\qquad \square$

### A.4 Proof of Proposition 4.6

*Proof.* From Proposition 4.4, we determine the form. Otherwise, if $x_\beta < x_\gamma$, we cannot draw conclusions. $\square$

### A.5 Proof of Theorem 4.7

*Proof.* Consider the sequence of steps required to make $\mathbf{x}'$ from $\mathbf{x}$,

$$\mathbf{x}^1 \to \mathbf{x}^2 \to \cdots \to \mathbf{x}',$$

with $\mathbf{x}^1 = \mathbf{x}$. For each step, we can either reduce the value of $x_i$ as long as $x_i' \geq 0$, or swap $x_i$ with $x_j$ if $x_i > x_j$. We would like to construct a new sequence by applying reduction operations and swap operations one by one. In order to accomplish this, we merge all reduction operations between two swap operations. In the absence of a reduction operation, we simply consider $\psi(\mathbf{x}, \mathbf{0})$. As a result, we have

$$\mathbf{x}^{i+1} = \begin{cases} \psi(\mathbf{x}^i, \mathbf{c}^i), & \text{if } i \text{ odd}, \\ \Gamma(\mathbf{x}^i, u^i, v^i), & \text{if } i \text{ even}. \end{cases} \tag{29}$$

Next, we construct another sequence in $\partial\Omega$ to bound $\mathbf{x}^i$,

$$\widetilde{\mathbf{x}}^1 \to \widetilde{\mathbf{x}}^2 \to \cdots \to \widetilde{\mathbf{x}}',$$

with $\widetilde{\mathbf{x}}^1 = \mathbf{x}$ and

$$\widetilde{\mathbf{x}}^{i+1} = \begin{cases} \widetilde{\mathbf{x}}^i, & \text{if } i \text{ odd}, \\ \Gamma(\widetilde{\mathbf{x}}^i, u^i, v^i) & \text{if } i \text{ even and } \widetilde{\mathbf{x}}^i_{u^i} > \widetilde{\mathbf{x}}^i_{v^i}, \\ \widetilde{\mathbf{x}}^i, & \text{if } i \text{ even and } \widetilde{\mathbf{x}}^i_{u^i} < \widetilde{\mathbf{x}}^i_{v^i}. \end{cases}$$

We want to show that $\widetilde{\mathbf{x}}^i \geq \mathbf{x}^i$ for all $i$. It is clear that this holds for $i = 1$, and we consider when $i > 1$. We focus on the third case because the first two cases are obvious. If $\widetilde{\mathbf{x}}^i_{u^i} < \widetilde{\mathbf{x}}^i_{v^i}$ and $\mathbf{x}^i_{u^i} > \mathbf{x}^i_{v^i}$, since $\mathbf{x}^i_{u^i} \leq \mathbf{x}^{i-1}_{u^{i-1}}$ and $\mathbf{x}^{i-1}_{u^{i-1}} \leq \widetilde{\mathbf{x}}^i_{u^i}$, then $\mathbf{x}^i_{u^i} < \widetilde{\mathbf{x}}^i_{v^i}$ and $\mathbf{x}^i_{v^i} < \widetilde{\mathbf{x}}^i_{u^i}$. Thus, after swapping on $\mathbf{x}^i$, $\mathbf{x}^{i+1} \leq \widetilde{\mathbf{x}}^{i+1}$. By induction, we conclude.

$\square$

### A.6 Proof of Proposition 4.11

*Proof.* Let $c = x_\beta - x_\beta'$, then we have

$$f(x_\beta', x_\gamma', \mathbf{x}_\neg) = f(x_\beta', x_\gamma + c, \mathbf{x}_\neg) \underset{M}{\leq} f(x_\beta' + c, x_\gamma, \mathbf{x}_\neg) = f(x_\beta, x_\gamma, \mathbf{x}_\neg).$$

$\square$

### A.7 Proof of Theorem 4.12

*Proof.* First, we show if $\mathbf{x}' \in \Omega(\mathbf{x})$, then $f(\mathbf{x}') \underset{M}{\leq} f(\mathbf{x})$. Denote $c_i = x_i - x_i'$, then from Equation equation 20 we have

$$\sum_{j=1}^{i} c_j \geq 0, i = 1, \ldots, m.$$

By Proposition 4.11 and individual monotonicity, we have

$$
\begin{aligned}
f(x_1, \ldots, x_m) &\underset{M}{\geq} f(x_1', x_2 + c_1, \ldots, x_m) \\
&\underset{M}{\geq} f(x_1', x_2', x_3 + c_1 + c_2, \ldots, x_m) \\
&\underset{M}{\geq} \cdots \\
&\underset{M}{\geq} f\left(x_1', \ldots, x_m' + \sum_{i=1}^{m} c_i\right) \\
&\underset{M}{\geq} f(x_1', \ldots, x_m').
\end{aligned}
$$

Conversely, suppose $\mathbf{x}' \notin \Omega(\mathbf{x})$, then $\exists i$ such that $\sum_{j=1}^{i} x_j < \sum_{j=1}^{i} x_j'$. Let $c = \sum_{j=1}^{i} x_j$. Consider the function

$$
f(x_1, \ldots, x_m) = 1_{\sum_{j=1}^{i} x_j > c}.
$$

Clearly, $f$ satisfies the individual and strong pairwise monotonicity. However, we have

$$
0 = f(\mathbf{x}) \underset{M}{\leq} f(\mathbf{x}') = 1.
$$

Thus, we conclude.

$\square$

### A.8 Proof of Proposition 5.1

*Proof.* By individual monotonicity, we know if $\mathbf{x}' \in \phi(\mathbf{x})$, then $\widehat{\mu}(\mathbf{x}') \underset{M}{\leq} \widehat{\mu}(\mathbf{x})$. $\square$

### A.9 Proof of Proposition 5.2

*Proof.* If there is $\mathbf{x}' \in \Omega(\mathbf{x})$ has not been explored, then $\mathbf{x}' + \mathbf{e}_i$ has not been explored for all $i$ except at boundaries. By Proposition 4.6, Theorem 4.7 and Proposition 4.11, we know $\mathbf{x}' \leq \widetilde{\mathbf{x}}$ for some $\widetilde{\mathbf{x}} \in \partial\Omega(\mathbf{x})$ and all maximum boundary points are included in the initial list. It is possible to reach max boundary points if we continue adding $\mathbf{e}_i$ for some $i$, as a contradiction. $\square$

## B  Monotonic Groves of Neural Additive Models

[c1]Here, we briefly review the Monotonic Groves of Neural Additive Models (MGNAMs) introduced by Chen & Ye (2023). [c2]Assume that the features are divided into individual monotonic, weak pairwise monotonic, strong pairwise monotonic, and nonmonotonic parts, as $\mathbf{x} = (\mathbf{x}_S, \mathbf{x}_U, \mathbf{x}_P, \mathbf{x}_\neg)$, similar to Section 4. [c3] For features with weak pairwise monotonicity in $U$, we give two lists $\mathbf{u}$ and $\mathbf{v}$ with $|\mathbf{u}| = |\mathbf{v}|$ such that $f$ is weakly pairwise monotonic with respect to $x_{u_i}$ over $x_{v_i}$ for $i = 1, \ldots, |\mathbf{u}|$. For strong pairwise monotonicity, we assume that there is a list $\mathbf{p}$ such that $f$ is strongly pairwise monotonic to $x_{p_j}$ over $x_{p_{j+1}}$ for $j = 1, \ldots, |\mathbf{p}| - 1$. All monotonic features follow individual monotonicity. GMNAMs use a special architecture of

$$
f(\mathbf{x}; \boldsymbol{\Theta}) = \alpha + \sum_{s:s \in S} f(x_s; \boldsymbol{\theta}_s) + \sum_{u:u \in U} f(x_u; \boldsymbol{\theta}_u) + f(\mathbf{x}_P; \boldsymbol{\theta}_P) + \sum_{\gamma \in \neg} f(x_\gamma, \boldsymbol{\theta}_\gamma),
$$

[c4]where each $f$ is parametrized by neural networks, $\boldsymbol{\theta}_s, \boldsymbol{\theta}_u, \boldsymbol{\theta}_P, \boldsymbol{\theta}_\gamma$ are parameters for each individual neural networks, and $\boldsymbol{\Theta}$ include all parameters. The architecture is assumed to keep the model transparent, where

---

[c1] *Text added.*
[c2] *Text added.*
[c3] *Text added.*
[c4] *Text added.*

$f(x, \boldsymbol{\theta}_s), f(x, \boldsymbol{\theta}_u), f(x, \boldsymbol{\theta}_\gamma)$ only takes one feature and $f(\mathbf{x}, \boldsymbol{\theta}_P)$ is a function of several features with strong pairwise monotonicity. This architecture is assumed since if the function is additively separated, then strong pairwise monotonic features have difficulty exhibiting diminishing marginal effects, which is common in social science. Following this, the standard regularization methods are applied

$$\min_{\boldsymbol{\Theta}} \ell(\boldsymbol{\Theta}) + \lambda_1 h_1(\boldsymbol{\Theta}) + \lambda_2 h_2(\boldsymbol{\Theta}) + \lambda_3 h_3(\boldsymbol{\Theta}), \tag{30}$$

[c1]where $\ell(\Theta)$ is the mean-squared error for regressions and log-likelihood function for classification, and $h_1, h_2, h_3$ are corresponding penalties for individual, weak pairwise, and strong pairwise monotonicity. Let us assume that all features are already binned with sets $\mathbb{S}_i$ for each $s \in S$, $\mathbb{U}_i$ for each $u \in U$, and $\mathbb{P}$. To simplify the notation, we assume that all features are binned equally with the distance $\Delta x$. As a result, we have

$$h_1(\boldsymbol{\Theta}) = \sum_{s \in S} \sum_{x_i \in \mathbb{S}_i} \max\left(0, f(x_i + \Delta x; \boldsymbol{\theta}_s) - f(x_i; \boldsymbol{\theta}_s)\right),$$

$$h_2(\boldsymbol{\Theta}) = \sum_{u \in U} \sum_{x_i \in \mathbb{U}_i} \max\left(0, f(x_i + \Delta x; \boldsymbol{\theta}_u) - f(x_i; \boldsymbol{\theta}_u)\right) + \sum_{j=1}^{|\mathbf{u}|} \sum_{x_i \in \mathbb{U}_i} \max(0, f(x_i; \boldsymbol{\theta}_{u_j}) - f(x_i; \boldsymbol{\theta}_{v_j})),$$

$$h_3(\boldsymbol{\Theta}) = \sum_{p \in P} \sum_{\mathbf{x}_i \in \mathbb{P}} \max(0, f(\mathbf{x}_i + \mathbf{e}_p; \boldsymbol{\theta}_P) - f(\mathbf{x}_i; \boldsymbol{\theta}_P)) + \sum_{j=1}^{|\mathbf{p}|-1} \sum_{\mathbf{x}_i \in \mathbb{P}} \max(0, f(\mathbf{x}_i + \mathbf{e}_{p_j}; \boldsymbol{\theta}_P) - f(\mathbf{x}_i + \mathbf{e}_{p_{j+1}}; \boldsymbol{\theta}_P)),$$

[c2] whereas $(\mathbf{e}_i)_j = \delta_{i,j}$, whereas $\delta_{i,j}$ is the kronecker delta. The model is trained by algorithm 2 .

---

**Algorithm 2** Monotonic Groves of Neural Additive Models (MGNAMs)

---

1: **Initialization**: $\lambda_1 = \lambda_2 = \lambda_3 = 0$, the architecture of the groves of neural additive models $(S, U, P, \neg)$
2: Train a groves of neural additive model by equation 30
3: **while** $\min(h_1, h_2, h_3) > 0$ **do**
4:     Increase $\lambda_i$ if $h_i > 0$
5:     Update the groves of neural additive models by equation 30.
6: **end while**

---

# C   DATA and MODELS

[c3]A summary of the results is presented in Table 7. [c4]Accuracy of models is evaluated by the AUC. We find that the AUC is almost the same before and after monotonicity is imposed, which is consistent with the findings of Chen & Ye (2023); Wang & Gupta (2020); Gupta et al. (2020). [c5]As a reminder, monotonicity is primarily intended for the purposes of conceptual soundness and fairness. Further, this paper does not focus on the accuracy of the model, but rather on the effectiveness of finding global maximums by equation 11. [c6]Reporting the results of accuracy is only for the purpose of completeness.

## C.1   Finance - Credit Scoring

### C.1.1   Data Description

We use the Kaggle credit score dataset [c6].

---

[c1] *Text added.*
[c2] *Text added.*
[c3] *Text added.*
[c4] *Text added.*
[c5] *Text added.*
[c6] *Text added.*
[c6]https://www.kaggle.com/c/GiveMeSomeCredit/overview

Table 7: Summary results for all datasets

| DATASETS | GMSC | COMPAS | LAW | LIFE-SCIENCE | MAMMOGRAPHY |
|---|---|---|---|---|---|
| AUC (%) | 85 | 72 | 86 | 68 | 90 |
| OOD (%) | 2.8 | 10.4 | 10.5 | 14.0 | 52.2 |

- $x_1 - x_3$: Last two years, the number of times borrower was 90+ days past due, 60-89 days past due, and 30-59 days past due.

- $x_4$: Monthly income.

- $x_5$: Number of dependents in the family.

- $x_6$: Total balance on credit cards and personal lines of credit except for real estate and no installment debt such as car loans divided by the sum of credit limits.

- $x_7$: Monthly debt payments, alimony, and living costs divided by monthly gross income.

- $x_8$: Number of open loans and lines of credit

- $x_9$: Number of mortgage and real estate loans

- $x_{10}$: Age of borrower in years.

- $y$: Client's behavior; 1 = Person experienced 90 days past due delinquency or worse.

We impose strong pairwise monotonicity of $x_1 - x_3$ and individual monotonicity for $x_4 - x_5$.

For simplicity, data with missing variables are removed. Past dues that are greater or equal to 20 are discarded. Then past dues greater than four times are replaced by four due to the rarity. This also applies to $x_5$ if its value exceeds five. To apply the fast marching method, we categorize $x_4$ into the following intervals: $[0, \$2500)$, $[\$2,500, \$5,000)$, $[\$5,000, \$7,500)$, $[\$7,500, \$10,000)$, $[\$10,000, \$50,000)$, and $[\$50,000, \infty)$. Afterward, they are transformed from five to zero so that $f$ increases monotonically with respect to $x_4$. We make such a choice in order to make features as easy to understand as possible for customers. This is not a unique choice. The model performance has been monitored to ensure that the accuracy does not deteriorate. When checking for accuracy, the dataset is randomly partitioned into 70% training and 30% test sets.

### C.1.2 Model

For MGNAM, we consider the architecture

$$f(\mathbf{x}) = f_{1,2,3}(x_1, x_2, x_3) + f_4(x_4) + \cdots + f_{10}(x_{10}). \tag{31}$$

In other words, $x_1 - x_3$ are grouped together, and the remaining features are handled using 1-dimensional functions. For $x_1 - x_3$, we enforce strong pairwise monotonicity. We enforce individual monotonicity for $x_4 - x_5$. All functions are approximated by neural networks with one hidden layer of four neurons. We focus on simple architectures since there is no apparent improvement in accuracy for more complicated models.

### C.1.3 Results

The area-under-the-curve (AUC) of the model is around 85%, which indicates that the model is accurate. It might be possible to improve model performance by further cleaning the data, but since this is not the primary concern of our study, we opt to omit it for simplicity.

### C.2 Criminology - Recidivism

#### C.2.1 Data Description

COMPAS is a proprietary score developed to predict recidivism risk, which is used to guide bail, sentencing, and parole decisions. A report published by ProPublica in 2016 provided recidivism data for defendants in Broward County, Florida (Pro, 2016). We focus on the simplified cleaned dataset provided in Dressel & Farid (2018). Three thousand and fifty-one (45%) of the 7,214 observations committed a crime within two years. This study uses a binary response variable, recidivism, as the response variable. The dataset here contains nine features selected after some feature selection was conducted.

- $x_1$: Total number of juvenile felony criminal charges

- $x_2$: Total number of juvenile misdemeanor criminal charges

- $x_3$: Age

- $x_4$: Total number of non-juvenile criminal charges

- $x_5$: A numeric value corresponding to the specific criminal charge

- $x_6$: An indicator of the degree of the charge: misdemeanor or felony

- $x_7$: Races include White (Caucasian), Black (African American), Hispanic, Asian, Native American, and Others

- $x_8$: Sex, male or female

- $x_9$: A numeric value between 1 and 10 corresponds to the recidivism risk score generated by COMPAS software (a small number corresponds to a low risk, and a larger number corresponds to a high risk)

- $y$: Whether the defendant recidivated two years after the previous charge

To avoid discrimination, we further exclude races and sexes. The COMPAS score is also excluded as it is not the focus of this study and is correlated with other features, making its interpretation more difficult. As there are too few samples, we truncate the number of juveniles exceeding five. Otherwise, if monotonicity is requested, neural network functions will become flat, which is not helpful.

#### C.2.2 Model

For MGNAM, we consider the architecture

$$f(\mathbf{x}) = f_{1,2}(x_1, x_2) + f_3(x_3) + \cdots + f_6(x_6). \tag{32}$$

In other words, $x_1 - x_2$ are grouped, and the remaining features are handled using 1-dimensional functions. For $x_1 - x_2$, we enforce strong pairwise monotonicity. We further impose individual monotonicity on $x_4$ and $x_6$.

#### C.2.3 Result

The AUC of the model is about 72%, which is consistent with the literature (Dressel & Farid, 2018).

We calculate the global feature importance by BShap in Figure 5. [c1]A brief description of BShap is provided in Section G. We take the mean value as the baseline value $\mathbf{x}'$. This result indicates that $x_3$, the Age, is an essential feature.

---

[c1] *Text added.*

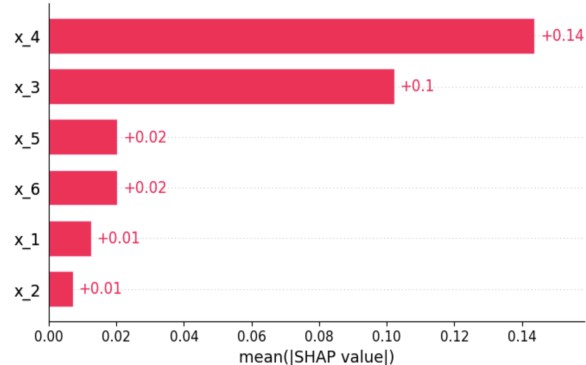

Figure 5: Global feature importance of COMPAS using BShap.

### C.3   Education - Law School Bar Exam

#### C.3.1   Data description

The law school dataset (Wightman, 1998) concerns information on the probability of passing the bar examination. In 1991, 163 law schools in the United States were surveyed by the Law School Admission Council (LSAC). From the total of 18,692 observations, 16,856 (90%) people passed the bar for the first time. If, for instance, universities wish to award scholarships based on the likelihood of passing the bar examination, fairness could be important. In this study, the response variable is a binary variable, pass. There are 11 features in this dataset.

- $x_1$: The student's decile in the school given his grades in Year 3

- $x_2$: The student's decile in the school given his grades in Year 1

- $x_3$: The student's LSAT score

- $x_4$: The student's undergraduate GPA

- $x_5$: Whether the student will work full-time or part-time

- $x_6$: The student's family income bracket

- $x_7$: Tier, which is an indicator of school quality

- $x_8$: Whether the student is a male or female

- $x_9$: Race

- $x_{10}$: The student's first-year law school GPA

- $x_{11}$: The student's cumulative law school GPA

- $y$: Whether the student passed the bar exam on the first try

Race and sex were excluded for potential bias. The law school GPA (LGPA) is calculated on different scales for the first year and the cumulative. To make a comparison, we scale them. $x_{10} - x_{11}$ are excluded as they are highly correlated with $x_1 - x_2$. Additionally, to avoid unfairness, gender and race are also excluded. Hence, the first 7 features remained to train the model.

Table 8: The comparison of the optimization 11 by considering different monotonicity for the Law school dataset with samples affected by pairwise monotonicity. Pairwise monotonicity improves the result.

| | **INDIVIDUAL** | | | **WEAK** | |
| $\tau$ | $\frac{|\mathbb{U}|}{|\mathbb{T}|}(\%)$ | MEAN-ITER | $\tau$ | $\frac{|\mathbb{U}|}{|\mathbb{T}|}(\%)$ | MEAN-ITER |
| 0.5 | 88.0 | 9 | 0.5 | 87.4 | 13 |
| 0.4 | 88.0 | 9 | 0.4 | 87.4 | 13 |
| 0.3 | 79.9 | 268 | 0.3 | 73.9 | 400 |
| 0.2 | 62.2 | 740 | 0.2 | 59.6 | 1220 |
| 0.1 | 39.5 | 985 | 0.1 | 38.9 | 1850 |
| 0 | 38.7 | 991 | 0 | 38.2 | 1862 |

### C.3.2   Model

For MGNAM, we consider the architecture

$$f(\mathbf{x}) = f_1(x_1) + f_2(x_2) + \cdots + f_7(x_7). \tag{33}$$

For all grade-related features $(x_1 - x_4)$, we require individual monotonicity, as well as weak pairwise monotonicity for $x_1$ over $x_2$. In the latter cases, the requirement indicates the pairwise monotonicity of time: the more recent information should be regarded as more valuable.

### C.3.3   Results

The AUC of the model is about 86%. Regarding the FMM results, approximately 86 percent of OOD data obtained a confident lower bound. The global BShap value is calculated in Figure 6. There is great significance to the $x_1$, $x_2$, and $x_7$ features. This model is designed to ensure fairness by not considering $x_7$, the tier of the law school, as a monotonic feature. However, the tier is an important feature that contributes to the uncertainty associated with the prediction. Taking the example of Figure 7, high variance data can be observed in cases where the law school's tier is high and the student's LSAT is low. Since most admitted students to top-tier schools possess high LSAT scores, this is intuitively reasonable. Without the monotonic information of tier, high-variance data cannot be effectively handled. To mitigate bias, monotonicity on tier should be avoided. If the feature tier could be replaced with other unbiased yet indicative monotonic features, such as historical bar exam passing rates of schools, we believe our performance could be further improved.

A comparison of individual monotonicity and pairwise monotonicity is presented in Table 8. The improvement has been observed.

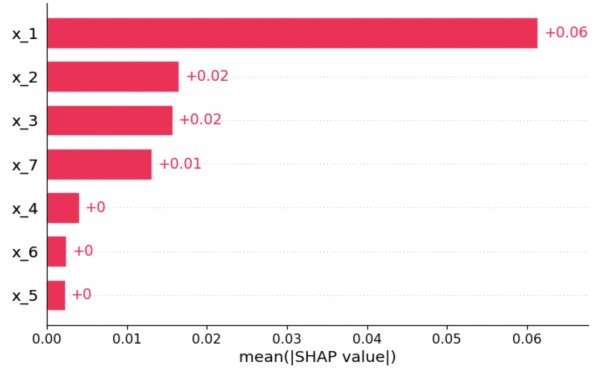

Figure 6: Global feature importance of LawSchool using BShap.

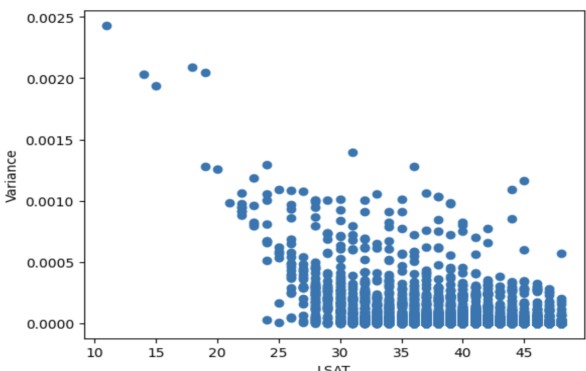

Figure 7: Variance vs LSAT in the Law School dataset.

## C.4   Life science - Happiness survey

This dataset [c0] is about the Somerville happiness survey by Koczkodaj (2018). The results of this study are used by the City to make decisions regarding the future development of Somerville. OOD datasets are commonly found in such surveys due to the fact that a large number of participants is unlikely. Moreover, our framework is well suited for this type of application since most of the features exhibit monotonic behavior based on domain knowledge. The following features are described.

### C.4.1   Data description

- $x_1$: The accessibility of information regarding city services

- $x_2$: The housing cost

- $x_3$: The overall quality of public schools

- $x_4$: People's confidence in the local police

- $x_5$: Upkeep of streets and sidewalks

- $x_6$: Presence of social community events

- $y$: Whether a person is happy or not

### C.4.2   Model

For MGNAM, we consider the architecture

$$f(\mathbf{x}) = f_1(x_1) + f_2(x_2) + \cdots + f_6(x_6). \tag{34}$$

We impose individual monotonicity on all features.

### C.4.3   Results

Our model has an AUC of approximately 68 percent. The prediction of happiness is noisy by nature, so we consider the result to be satisfactory. The performance of FMM is excellent since all features are monotonic.

---

[c0]https://archive.ics.uci.edu/dataset/479/somerville+happiness+survey

### C.5 Medical - Mammographic Mass

#### C.5.1 Data description

As a screening tool for breast cancer, mammography is widely used in the medical field. However, due to the uncertainty prediction, biopsies that proved to be benign are not examined as thoroughly as they should be. There has been some previous research relating to semi-supervised learning to reduce the uncertainty of a model by Calderon-Ramirez et al. (2021). On the other hand, our approach reduces the uncertainty of OOD data by finding a lower bound, which is based on the monotonicity property. The data for Mammographic Mass is collected by Elter (2007), for a 5-feature-based classification. Overall, 961 data are available, including 516 benign and 445 malignant. Below are illustrations of all features.

- $x_1$: BI-RADS assessment, which is a standard assessment used by doctors to describe mammograms. The values range from 1, the benign, to 5, with a high possibility of malignancy.

- $x_2$: Age

- $x_3$: Shape of Mammography, classified into four types: round, oval, lobular, and irregular

- $x_4$: Margin of Mammography, classified into circumscribed, microlobulated, obscured, ill-defined, and spiculated

- $x_5$: Mammographic density, classified as high, iso, low and fat-containing

- $y$: The binary label, malignancy=1, benign=0

#### C.5.2 Model

For MGNAM, we consider the architecture

$$f(\mathbf{x}) = f_1(x_1) + f_2(x_2) + \cdots + f_5(x_5). \tag{35}$$

According to the doctor's diagonalization, $x_1$ is monotonic for predicting the severity of breast cancer. Based on previous research, there is a highly positive relationship between mammographic density and cancer severity, as discussed in Nazari & Mukherjee (2018), therefore, we impose individual monotonicity on both $x_1$ and $x_5$.

#### C.5.3 Results

The AUC of the model is about 90%. The OOD data is more than half due to the rare samples. [c1]Therefore, one should be cautious when making predictions based on the model because it is quite uncertain about its predictions. If there is more data available in the hospital, this could be reduced. Based on FMM, we obtain lower bounds of confidence of approximately 30%, demonstrating the effectiveness of our approach. In the case of healthcare datasets, we believe that it is definitely possible to improve the performance by using more features as well as domain knowledge. Due to the fact that we are not experts in this field, we do not feel comfortable imposing domain knowledge in more complex situations. This is only a simple example provided for demonstration purposes.

## D Algorithm Details

### D.1 Details for MGNAMs

There are two steps in the training process for the model.

Initially, a mini-batch gradient descent approach with a batch size of 64 is applied to pre-train the model without taking into account monotonicity. The initial learning rate is $1 \times 10^{-2}$, and it is multiplied by 0.1

---

[c1] *Text added.*

when there is no improvement in the loss over 5 epochs, and early stopping is performed when there is no improvement over 10 epochs.

As part of Step 2, the model is trained to satisfy all monotonicity requirements and the batch gradient descent method is applied. The $\alpha$ factor, which represents the punishment for features exhibiting monotonicity violations, is initially set at $1 \times 10^{-1}$ and is multiplied by 10 every 10 epochs. At the same time, the learning rate at is set to $1 \times 10^{-2}$. Once all monotonies have been satisfied, the remuneration will change to $1 \times 10^{-3}$ for another 10 epochs of training.

### D.2 Details for Ipopt

[c1]Ipopt (Hart et al., 2017; 2011) [c2]is a software package for large-scale nonlinear optimization. It can be used to solve general nonlinear programming problems of the following type

$$\begin{cases} \min_{\mathbf{x} \in \mathbb{R}^n} & f(\mathbf{x}), \\ \text{s.t.} & \mathbf{g}^L \leq \mathbf{g}(\mathbf{x}) \leq \mathbf{g}^U, \\ & \mathbf{x}^L \leq \mathbf{x} \leq \mathbf{x}^U, \end{cases}$$

[c3]where $\mathbf{x}$ are the optimization variables, $f : \mathbb{R}^n \to \mathbb{R}$ is the objective function, and $\mathbf{g} : \mathbb{R}^n \to \mathbb{R}^m$ are the general nonlinear constraints. The functions $f$ and $\mathbf{g}$ can be linear or nonlinear and convex or non-convex (but should be twice continuously differentiable). Ipopt implements the interior point line search filter method to find a local solution to the nonlinear programming problem.

[c4] In the implementation, we maintain all parameters as default settings while changing the maximum number of iterations to 1000. For greater accuracy, we use a large maximum number of iterations. As the package ignores the potential discrete nature of features, we round the continuous numbers to integer numbers after the optimization. In the case of monotonically increasing features, we round them down; otherwise, we round them up.

## E  Other Discussions

### E.1  Binning

By binning or discretizing, continuous features are transformed into discrete ones. Binning is a common practice in numerical partial differential equations (Sethian, 1996). For ML, binning may improve predictive models' accuracy by reducing noise or nonlinearity in the dataset as well as identifying outliers, and invalid and missing values of numerical features. The use of bins is particularly popular in high-risk sectors, where interpretation is of paramount importance. When considering the income feature, for example, people are more likely to consider low-, median-, and high-income classes rather than specific figures. Various binning methods are available, including equal width, equal frequency, and weight of evidence. As the choice of methods depends heavily on the application and the appetite of the user, we will not discuss this further. It is possible to preserve continuous features by leaving original features alone and binning only the features in the optimization process.

### E.2  Choice of Models

We apply accurate and transparent monotonic groves of the neural additive model (MGNAM) proposed in Chen & Ye (2023), as three types of monotonicity are included. The code is modified based on the Neural Additive Models (Agarwal et al., 2021). In general, the choice is not unique. Models developed by Liu et al. (2020); Milani Fard et al. (2016); You et al. (2017); Runje & Shankaranarayana (2023) are applicable for individual monotonicity, whereas deep lattice models (Gupta et al., 2020; Cotter et al., 2019) include strong pairwise monotonicity.

---

[c1] *Text added.*
[c2] *Text added.*
[c3] *Text added.*
[c4] *Text added.*

Table 9: [c5]OOD results for different $\epsilon$

| DATASETS | GMSC | COMPAS | LAW | LIFE-SCIENCE | MAMMOGRAPHY |
|---|---|---|---|---|---|
| OOD (%), $\epsilon = 10^{-2}$ | $8 \times 10^{-3}$ | 0 | 0 | 0 | 6.5 |
| OOD (%), $\epsilon = 10^{-3}$ | 2.8 | 10.4 | 10.5 | 14.0 | 52.2 |
| OOD (%), $\epsilon = 10^{-4}$ | 17.8 | 95.8 | 41.8 | 100 | 90.0 |

Table 10: [c7]Additional results by the FMM using all monotonicity

| DATASETS | $\epsilon$ | $\frac{|\mathbb{V}|}{|\mathbb{S}|}$ (%) | | | | | | MEAN-ITER |
|---|---|---|---|---|---|---|---|---|
| | | $\tau = 0.5$ | 0.4 | 0.3 | 0.2 | 0.1 | 0 | |
| GMSC | $10^{-4}$ | 100.0 | 99.8 | 99.8 | 99.5 | 83.3 | 7.2 | 63 |
| MAMMOGRAPHY | $10^{-2}$ | 94.4 | 81.5 | 77.8 | 72.2 | 63.0 | 63.0 | 9 |

### E.3 Choice of the Ensemble size $M$ and Threshold $\epsilon$

[c5]Through extensive empirical studies on a variety of datasets, Ovadia et al. (2019) [c6]suggests an ensemble size of $M = 5$ or 10. To ensure more robust results, we use $M = 10$.

$\epsilon$ thresholds are not unique and depend on applications and user preferences. In high-risk sectors, one may choose a very small value for $\epsilon$. However, if the risks are tolerable, a larger $\epsilon$ may be chosen. The finance sector, for instance, has seen different types of investors, including risk-averse, risk-neutral, and risk-seeking investors. There is a discussion in Section 3.6 by Petters & Dong (2016).

[c1]In this paper, we focus on the choice $\epsilon = 10^{-3}$ such that it works better for all models. We have also tried different values of $\epsilon$. The results for the number of OOD data are recorded in Table 9. [c2]When $\epsilon$ is very small, given the size of the dataset, the model could be uncertain about a large portion of the dataset, thus making the whole model uncertain. If $\epsilon$ is too large, models are already very confident about almost all predictions. The OOD data may not need to be further processed in such extreme cases. As a result, we mainly report the case when $\epsilon = 10^{-3}$. [c3]We present some useful cases for other $\epsilon$. The result of GSMC is provided when $\epsilon = 10^{-4}$ since other models are too uncertain. In addition, we provide the results of Mammography when $\epsilon = 10^{-2}$ since other models are very confident. The results are presented in Table 10.

## F Fast Marching Methods for the Eikonal Equation

[c8]Originally, the FMM (Tsitsiklis, 1995; Sethian, 1996; Helmsen et al., 1996) [c9]was proposed to solve the Eikonal equation as follows:

$$|\nabla u(\mathbf{x})| = \frac{1}{f(\mathbf{x})}, \text{ for } \mathbf{x} \in \Omega,$$
$$u(\mathbf{x}) = 0, \text{ for } \mathbf{x} \in \partial\Omega,$$

[c10]where $|\cdot|$ is the Euclidean norm, $\nabla$ is the gradient, $f$ is given, $\Omega$ is the domain, and $\partial\Omega$ is the boundary of the domain.

---

[c5] *Text added.*
[c6] *Text added.*
[c1] *Text added.*
[c2] *Text added.*
[c3] *Text added.*
[c8] *Text added.*
[c9] *Text added.*
[c10] *Text added.*

[c11]The following example illustrates how FMM was used in a two-dimensional setting. Assume that the domain has been discretized into a mesh. Meshpoints will be referred to as nodes. Every node $(x_i, y_j)$ has a corresponding value $U_{i,j} = U(x_i, y_j) \approx u(x_i, y_j)$. There are three sets in the algorithm, which are the far set, the considered set, and the accepted set. The far set includes the points that have yet to be calculated, the considered set includes the points that have already been calculated, but do not have the satisfactory solution, and the accepted set contains the points that have the desired solution. Briefly, the algorithm consists of the following steps:

1. [c1]In the initialization, assign every node $(x_i, y_j)$ the value of $U_{i,j} = +\infty$ and label them as far; for all nodes $(x_i, y_j) \in \partial\Omega$, set $U_{i,j} = 0$ and label $(x_i, y_j)$ as accepted.

2. [c2]For every far node $(x_i, y_j)$, calculate the new value for $\widetilde{U}$ using Eikonal's update formula. If $\widetilde{U} < U_{i,j}$, then set $U_{i,j} = \widetilde{U}$ and label $(x_i, y_j)$ as considered. To be more specific, for the first-order approximation, we have

$$\max\left(D_{i,j}^{-x}U, -D_{i,j}^{+x}U, 0\right)^2 + \max\left(D_{i,j}^{-y}U, -D_{i,j}^{+y}U, 0\right)^2 = \frac{1}{f_{i,j}^2},$$

   [c3]where

$$D_{i,j}^{\pm x}U = \frac{U_{i\pm1,j} - U_{i,j}}{\pm\Delta x},$$
$$D_{i,j}^{\pm y}U = \frac{U_{i,j\pm1} - U_{i,j}}{\pm\Delta y}.$$

   [c4]Let

$$U_X = \min\left(U_{i-1,j}, U_{i+1,j}\right),$$
$$U_Y = \min\left(U_{i,j-1}, U_{i,j+1}\right).$$

   [c5]For sufficient small step size $\Delta x, \Delta y$ such that $\left|\frac{U_X}{\Delta x} - \frac{U_Y}{\Delta y}\right| \leq \frac{1}{f_{i,j}}$, we have the solution to the following quadratic equation

$$\left(\frac{U_{i,j} - U_X}{\Delta x}\right)^2 + \left(\frac{U_{i,j} - U_Y}{\Delta y}\right)^2 = \frac{1}{f_{i,j}^2}.$$

   [c6] It can be written as $aU_{i,j}^2 + bU_{i,j} + c$. The following solution is used due to the physical characteristics of the problem

$$\widetilde{U} = \frac{-b + \sqrt{b^2 - 4ac}}{2a}.$$

3. [c7]Let $(\widetilde{x}_i, \widetilde{y}_j)$ be the node with the smallest value $U$ in the considered set. Label $(\widetilde{x}_i, \widetilde{y}_j)$ as accepted and remove it from the considered set. **The FMM is utilized here. FMM determines the smallest value by using the heap sort.**

4. [c8] For each neighbor $(x_i, y_j)$ of $(\widetilde{x}_i, \widetilde{y}_j)$ that is not accepted, calculate a tentative value $\widetilde{U}$.

---

[c11] *Text added.*
[c1] *Text added.*
[c2] *Text added.*
[c3] *Text added.*
[c4] *Text added.*
[c5] *Text added.*
[c6] *Text added.*
[c7] *Text added.*
[c8] *Text added.*

5. [c9]If $\widetilde{U} < U_{i,j}$, then set $U_{i,j} = \widetilde{U}$. If $(x_i, y_j)$ was previously labeled as far, update the label to be considered.

6. [c10]Return to Step 3 if there is a considered node. If not, terminate the process.

[c1] In summary, as a result of Eikonal's update formula, the FMM selects the smallest value in each step, resulting in a monotonically nondecreasing sequence as the solution, which is similar to equation 24. [c2]The heap algorithm is used to increase the speed of determining the smallest value and removing it from the set.

## G   Shapley Value

[c3]Following Lundstrom et al. (2022), [c4]we call the point of interest $\mathbf{x}$ to explain as an explicand and $\mathbf{x}'$ a baseline. [c5]The Shapley value (Shapley et al. (1953)) [c6]takes as input a set function $v : 2^N \to \mathbb{R}$, which produces attributions $s_i$ for each player $i \in N$ that add up to $v(N)$.

**Definition G.1** (Shapley value). *The Shapley value of a player $i$ is given by:*

$$s_i = \sum_{S \subseteq N \setminus i} \frac{|S|!(|N| - |S| - 1)!}{|N|!} (v(S \cup i) - v(S)).$$

[c7]We focus on the Baseline Shapley (BShap) Sundararajan & Najmi (2020), in which

$$v(S) = f(\overline{\mathbf{x}}_S; \mathbf{x}'_{N \setminus S}).$$

[c8]That is, baseline values replace the feature's absence. We denote BShap attribution by $\mathrm{BS}_i(\mathbf{x}, \mathbf{x}', f)$ and $\mathrm{BS}_i$ sometimes. For example, suppose $f(x_1, x_2) = x_1 + x_2$, $\mathbf{x} = (x_1, x_2)$, $\mathbf{x}' = (0, 0)$, and $S = \{1\}$, then we have $v(S) = f(x_1, 0)$. One common choice of $\mathbf{x}'$ is to take the average of all samples. For the global feature importance of the $i$th feature, we take the average of the absolute values of BShap

$$\mathcal{A}_i = \frac{1}{n} \sum_{j=1}^{n} |\mathrm{BS}_i(\mathbf{x}_j, \mathbf{x}', f)|.$$

---

[c9] *Text added.*
[c10] *Text added.*
[c1] *Text added.*
[c2] *Text added.*
[c3] *Text added.*
[c4] *Text added.*
[c5] *Text added.*
[c6] *Text added.*
[c7] *Text added.*
[c8] *Text added.*

