# OpenReview forum: "Towards Certainty: Exploiting Monotonicity with Fast Marching Methods to Reduce Predictive Uncertainty"
_TMLR — Rejected by TMLR_

### Review · Reviewer_eJ6v · 2024-03-15

**Summary Of Contributions:**

Broadly speaking, the authors of this paper are interested in the "confidence" of decisions made by machine learning models and algorithms. I had a very difficult time trying to understand what the actual problem of interest in this paper is, and it is thus difficult to write a summary of the key contributions with much confidence. Below, I will try to briefly summarize the main points that seem to be underlying the authors' motivations, and at the same time highlight where I am tripping up.

- The first problem raised: machine learning models such as neural networks are often *"overly optimistic"* in their predictions, despite those predictions being *"out of distribution (OOD)"*, i.e., points that the predictor may likely get wrong. This is a natural issue, well-known in the context of "model calibration" (confidence calibration, etc.).

- Related to the above point, the authors refer to past work on *"detection of OOD data"*. With this context in place, the authors are interested in the possibility of learning from what we don't know; I take "what we don't know" to refer to points that are determined to be OOD. That's fine, but I have no idea how this basic motivation connects to the rest of the paper. The core formal "task" defined in this paper (equation (10)) asks that we find a input vector $\\mathbf{x}^{\\prime}$ which is *not* OOD (i.e., satisfies the $\\epsilon$-confident condition in terms of variance of ensemble model predictions). So the input space is divided into sets of varying degrees of "certainty" based upon some ensemble model assumed to be given from the start; then what? Simply assigning a confidence label to different inputs does not aid us in "learning from what we don't know." It is this gap that really troubled me.

I roughly understand what is being described in equation (10); the authors want to find $\\mathbf{x}^{\\prime}$ for which the ensemble is sufficiently confident, both in terms of average confidence level and agreement between the models. While I understand what is written, I cannot understand *why* the authors are considering this optimization task; furthermore, in the definition of $\\Omega(\\mathbf{x})$, the function $f$ is used. Isn't this $f$ supposed to be unknown?

Getting back to the main claims of this paper, the authors define a procedure (using ancillary OOD detector and equation (10)) which assigns confidence levels to input vectors; this is described in section 3, and centers around the optimization task defined in equation (10); I described various points of confusion surrounding (10) a couple paragraphs back. They position this as generalizing a similar procedure presented by Chen (2022), where generalization is in terms of the "monotonicity" used (pairwise versus individual); I presume this is supposed to be encoded in the definition of $\\Omega$, but the definition of inequality by monotonicity is unclear to me (and how it related to individual/pairwise notions), so the precise difference with previous literature was also not clear to me.

On top of this generalization, since the underlying problem has difficult constraints and need not be convex, the authors suggest using a global search algorithm called "fast marching method" where the effective search space is reduced using structural knowledge of monotonicity properties in the underlying predictor. They also put this procedure to work in some experimental tests, though they do not highlight the empirical findings or insights as one of their main contributions at the end of section 1.

**Audience:**

No

**Broader Impact Concerns:**

No concerns.

**Claims And Evidence:**

No

**Requested Changes:**

In the previous fields I've already highlighted several issues I ran into when reading and trying to parse the contents of this paper. Below, I will highlight several varied points I tripped up on while reading the paper, in no particular order.

- Intro: *"Office of the Comptroller of the Currency"*... what is this? Perhaps readers who are residents of the USA may be familiar with such a term, but TMLR is meant to be read by a global english-speaking audience. It's fine to use this as an example, but the term shouldn't be used as if it is common knowledge.

- Related to the previous point, the term *"past dues"* (page 1) is in my mind very unclear. An "applicant" has "past dues". What is past due? Library books? What is he/she applying for? Presumably a loan of money, but this is not said anywhere.

- Page 2: *"\\ldots otherwise the model would be unfair."* This is really a subjective and vague statement.

- The *"three major contributions"* on page 2 could use some refining. For example: by introducing pairwise monotonicity, we get that *"tighter bounds can be provided."* What does this mean? I really had trouble deciphering what the authors are interested in *bounding*.

- Prerequisites: equations (1) and (2) give the basic data generating process. The authors give (1) for regression and (2) for classification. Is this degree of generality actually valid? The whole idea of $f$ representing some kind of "confidence" score doesn't really make sense in the regression setting. In addition, $\\epsilon\_{i}$ isn't even defined, so (1) is basically meaningless and just confusing. I will also mention that the relation of $\\mathbf{x}\_{i}$ to $x\_{j} > 0$ is not explained when these symbols first appear (final lines of page 2).

- Section 2.2: we have $f\_{i}(\\cdot;\\boldsymbol{\\theta}\_{i})$ for the $i$th model candidate used in the ensemble, but $\\boldsymbol{\\theta}\_{i}$ is undefined and meaningless, plus having $i$ subscript on both $f\_{i}$ and $\\boldsymbol{\\theta}\_{i}$ is redundant and confusing.

- Theorem 3.1: I have no idea what "achieves all monotonicity" means. A convex combination of monotontic functions is obviously monotonic; I'm not sure what the authors are trying to get at with this result.

- Defn 3.2: I know this is perhaps the most critical definition in the paper, but I have no idea what *"inequality by monotonicity"* means. The authors explain this as knowing $f(\\mathbf{x}^{\\prime}) \leq f(\\mathbf{x})$ from *"domain knowledge (not from function outputs)", but I really cannot parse this. Since this definition is basically impenetrable for me, it really makes it hard to get into the rest of the analysis. I will also comment (again) here that $f$ is used in the definition of $\\Omega(\\cdot)$. Is this okay? In my mind, $f$ is an ideal quantity unknown by the learning algorithm.

- In equations (10) and (11) the authors distinguish between points that are "decided" and those that are "undecided", where the latter comes from having high variance OR not being sufficiently ($\\gamma$-) confident. Wasn't the original motivating problem of interest *overconfidence* of machine learning models on OOD points? All the effort in this paper seems to be centered on finding input points where an ensemble is sufficiently confident, but actually dealing with over-confident points is entirely abstracted away. Again, a gap between the initial problem of interest and what is actually being done here.

I could continue in the same fashion, but I think my overall point is clear. TMLR does not require submissions to have outstanding significance or novelty, but the exposition needs to be clear, and the main claims need to be solid. I think the present submission needs a fair bit of work (and a bit of re-thinking) to satisfy this.

**Strengths And Weaknesses:**

__Strengths:__

The broad problem of interest (how to deal with over-confidence and under-confidence in machine learning) is important, and the authors have made a genuine effort in designing a procedure which could potentially aid users of machine learning tools by highlighting degrees of uncertainty, in a setting that captures a more flexible notion of monotonicity than was covered in the previous literature.


__Weaknesses:__

Put concisely, I think the clarity and quality of exposition in this paper is severely lacking. The critical notions of monotonicity are not in my opinion defined clearly, nor is the ultimate goal of this research. A technical goal (in the form of equation (10)) is given, and a procedure to solve this is derived and analyzed, but the reader is forced to reverse-engineer the ultimate goal from what really amounts to a sub-routine in my mind, since it is not clearly stated. Say we could solve equation (10) perfectly with no effort. How would that help us?

---

### Review · Reviewer_b6XK · 2024-03-26

**Summary Of Contributions:**

The paper studies algorithms designed to extract more information from uncertain predictions by exploiting monotonicity. Specifically, the authors propose solving a mean-variance optimization problem, as introduced by Chen (2022), but allow two other types of monotonicity constraints. This approach leads to a non-convex mixed-integer nonlinear programming problem. By leveraging the monotonicity property, the authors use the fast marching method to find the global solution to the problem. Additionally, they discuss the monotonicity-induced geometry of the domain for different constraints. Experiments are conducted across multiple applications, including finance, criminology, education, life sciences, and healthcare. The results demonstrate that it is possible to provide confident bounds for a large portion of uncertain predictions through monotonicity.

**Audience:**

Yes

**Broader Impact Concerns:**

N/A.

**Claims And Evidence:**

No

**Requested Changes:**

- The differences in contributions and the overlaps between this paper and those by Chen (2022) and Chen & Ye (2023) are not clearly stated. The authors write "As an example, Chen (2022) does not consider the discrete nature of features and non-convexity.". Please gvie more details to better evaluate your contributions. Additionally, discuss how the experiments conducted in this paper differ from those in the mentioned works.

- Regarding the experiments, the authors fail to include baseline comparisons, focusing solely on their own method. This omission is a significant flaw, as it hinders evaluation against state-of-the-art benchmarks. For instance, what are the results without monotonicity constraints?

- The paper is missing a thorough discussion of the state-of-the-art on monotonic machine learning models for uncertainty prediction, as well as out-of-distribution detection.


- The Fast Marching Method (FMM) is well-known, yet the authors state, 'As a result, we develop the marching method,' and introduce Algorithm 1. Subsequently, they note, 'Unlike the original FMM, our approach incorporates general monotonicity derived from domain knowledge.' This is not clear. Please clearly differentiate your adapted algorithm from the standard FMM. Additionally, in Section 5.1, the acronym 'MM' is introduced without explanation. How does MM relate to FMM?"

The authors also write "By utilizing general types of monotonicity, we extend the FMM to solve our optimization problem". In what sense did you extend the FMM algorithm? Are you applying the FMM algorithm?

- The paper does not clearly justify or present the evaluation metrics used. Is a higher AUC the main target? What is the AUC of models without monotonicity constraints?

- The discussion of the unsuccessful example and the less satisfactory results is somewhat brief. Providing more details and analysis on these examples is essential to gain a deeper understanding of the proposed method's limitations.


- Section 4 is challenging to navigate. It would benefit from a clearer flow that guides the reader through the usefulness of each definition and proposition.

- The authors should introduce the concept of monotonicity, especially its role in managing uncertainty, early in the introduction. As it stands, the meaning of monotonicity remains unclear until the reader reaches the later sections.

- What do the authors mean by:
	- "we are still unable to come up with a satisfactory bound"? Unsatisfactory in what sense?
	- "resulting in tighter bounds, but also complicating the problem.". In what sense?
	- "but the FMM accounts for discrete features and offers global solutions.". What do you mean by global solutions? Do you mean a global maximum?


- For Mamography, 52.2% of the data points are OOD for only 961 datapoints. The authors write "The OOD data is more than half due to the rare samples.". The implications of such as large number of OOD points is not clear.

- No ablation study is shown for the hyperparameters M and epsilon. The authors use M = 10 and epsilon = 10^−3. The short paragraph in Appendix D.3 is not enough.

- The authors write "- We calculate the global feature importance by Shap in Figure 4.". Shap was not properly introduced in the paper.

- The authors write "FMM has a mean number of iterations of 29 for each optimization, demonstrating its efficacy.". Why do you consider 29 iterations as efficient?

- I suggest the authors to give a short presentaiton of the MGNAM model (more than a reference).

- Multiple symbols are not properly introduced at the beginning of Section 2 and should be clarified:
    - What do \(\epsilon_i\), \(y_i\), \(x_i\), and \(f\) represent? Which variables are random?
    - What does the index \(i\) signify? Is \(i = 1, \ldots, n\)?
    - How does \(\mathbf{x}_i\) differ from \(x_j\)?
    - What is \(f_i\), and how does it differ from the index \(i\) in \(x_i\)?


- The title 'When predictions are uncertain, can monotonicity help?' might be refined to more accurately mirror the specific content of the paper.


Minor comments:

- The symbol \leq_M is used in the proof of Theorem 3.1. but is only introduced in (9).

- Theorem 3.1 would be more appropriately named as a Proposition.

- Please improve these sentences:
	- "Lakshminarayanan et al. (2017) describes a simple yet effective approach to detecting OOD data and has shown to be the best performer by Ovadia et al. (2019). Ensemble methods are used."
	- "Models are then used as predictions based on their average,"
	- In the introduction, the sentence "Consequently, pairwise monotonicity improves the results" is not clear. What is the link with the previous sentence?

- In Figure 2, the font size of the axes labels is too small.

- In Expressions (16) and (17), please define the symbol "|".

- Typos and grammar
	- "and the results have shown promising"
	- "monotonic increasing functions." -> monotonically
	- Expression (3), the last alpha should not be in bold font
	- "the optimization problem equation 10"
	- "we want to explore its nearby."
	- "there are only 20% of which, find the confident lower bound"

**Strengths And Weaknesses:**

### Strengths

- The paper contributes to domain-knowledge-inspired machine learning which is an important topic in many applications.
- The paper generalizes the two-stage framework by Chen (2022) to other types of monotonicity.
- Experiments are performed on five different datasets from different domain applications.

### Weaknesses

- A major limitation of the paper is its clarity, with noticeable issues in notation and flow suggesting rushed writing (see details below).

- The differences in contributions and the overlaps between this paper and those by Chen (2022) and Chen & Ye (2023) are not clearly stated. The authors write "As an example, Chen (2022) does not consider the discrete nature of features and non-convexity.". Please gvie more details to better evaluate your contributions. Additionally, discuss how the experiments conducted in this paper differ from those in the mentioned works.


- Regarding the experiments, the authors fail to include baseline comparisons, focusing solely on their own method. This omission is a significant flaw, as it hinders evaluation against state-of-the-art benchmarks. For instance, what are the results without monotonicity constraints?

- The paper is missing a thorough discussion of the state-of-the-art on monotonic machine learning models for uncertainty prediction, as well as out-of-distribution detection.

---

### Review · Reviewer_piVt · 2024-04-12

**Summary Of Contributions:**

Monotonicity information of a response with respect to some variables may be available. The authors propose to exploit it to post-process predictions that have been deemed unconfident. That is, for a given element x whose prediction is deemed unconfident, if there is an element x’ whose prediction is confident while bounding the one at x by monotonicity, then this later can be used for decision. A corresponding methodology to identify such x’ is proposed, relying on a fast marching method. Examples are provided, showing improvements.

**Audience:**

Yes

**Claims And Evidence:**

No

**Requested Changes:**

The title is not precise enough on what the paper is really about.

Page 4: It is not clear why small values of $\hat{\mu}$ are not useful in a general context (other than credit scoring). In fact the mean-variance optimization problem is not motivated, in particular for general applications.

The choice of the fast marching method over possible alternatives is not discussed.

If the predictions follow the monotonicity properties, it is not clear what the proposed method actually adds. Perhaps the issue is that the confidence is only measured by the variance. For instance, if all f_i(x) are large, then the decision should be easy even if the variance is large.

P8: we want to explore its nearby. ?

Tables: mark best values

**Strengths And Weaknesses:**

Strengths
- considering pairwise monotonicity extends the univariate version, but also complexifies the setup.

Weaknesses
- the work is an extension of Chen (2022), adding pairwise monotonicity.
- the presentation and example focus on credit scoring, in a context with out-of-distribution data. This is somehow limiting.
- no alternative baseline is proposed, e.g., those proposed in D.2.

---

### Decision · Action_Editor_yMhB · 2024-05-17

**Recommendation:** Reject

**Comment:**

This manuscript concerns improving the predictive performance and uncertainty of neural networks by teaching them better about "what they don't know." To this end, the authors outline an optimization-based methodology exploiting various notions of monotonicity in an attempt to improve predictive uncertainty.

Although the reviewers agree that this material is of interest to TMLR's audience, they also identified several perceived weaknesses with the manuscript as submitted. namely:

- concerns regarding the clarity of the manuscript and some key ideas within,
- concerns regarding the design of the empirical study,
- a lack of adequate discussion of related work, in particular Chen (2022).

Ultimately, following the author-reviewer discussion period and some revisions by the authors, these issues were not completely resolved and deemed too serious to justify publication of the submitted/revised manuscript.

However, the reviewers also agree that the manuscript might be appropriate for TMLR if the issues above -- especially regarding clarity -- could be resolved in a resubmitted manuscript.

**Audience:**

Yes, the manuscript considers methods for improving the predictive performance of neural networks, which is of interest to a large subset of TMLR's audience.

**Claims And Evidence:**

No, the reviewers did not reach consensus that the claims made in this submission are supported by clear and convincing evidence. In particular, a majority of the reviewers take exception with the "clear" component of this acceptance criterion, feeling that the manuscript as submitted (and revised) is lacking in clarity regarding critical details.

**Resubmission Of Major Revision:**

The authors may consider submitting a major revision at a later time.